# DeepfakeBench-MM: A Comprehensive Benchmark for Multimodal Deepfake Detection

## Abstract

The misuse of advanced generative AI models has resulted in the widespread proliferation of falsified data, particularly forged human-centric audiovisual content, which poses substantial societal risks (*e.g.*, financial fraud and social instability). In response to this growing threat, several works have preliminarily explored countermeasures. However, the lack of sufficient and diverse training data, along with the absence of a standardized benchmark, hinder deeper exploration. To address this challenge, we first build **Mega-MMDF**, a large-scale, diverse, and high-quality dataset for multimodal deepfake detection. Specifically, we employ 21 forgery pipelines through the combination of 10 audio forgery methods, 12 visual forgery methods, and 6 audio-driven face reenactment methods. Mega-MMDF currently contains 0.1 million real samples and 1.1 million forged samples, making it one of the largest and most diverse multimodal deepfake datasets, with plans for continuous expansion. Building on it, we present **DeepfakeBench-MM**, the first unified benchmark for multimodal deepfake detection. It establishes standardized protocols across the entire detection pipeline and serves as a versatile platform for evaluating existing methods as well as exploring novel approaches. DeepfakeBench-MM currently supports 5 datasets and 11 multimodal deepfake detectors. Furthermore, our comprehensive evaluations and in-depth analyses uncover several key findings from multiple perspectives (*e.g.*, augmentation, stacked forgery). We believe that DeepfakeBench-MM, together with our large-scale Mega-MMDF, will serve as foundational infrastructures for advancing multimodal deepfake detection. Our DeepfakeBench-MM is released at https://github.com/AnonymousDeepfakeBench-MM/DeepfakeBench-MM for anonymous review. Research-only access to Mega-MMDF will be available after the review period.

## 1 Introduction

Deepfake technologies, which leverage advanced generative AI to synthesize realistic human-centric data (*e.g.*, voices (Casanova et al., 2024; 2022; Li et al., 2023), facial images/videos (Shiohara et al., 2023; Podell et al., 2024; Li et al., 2024b), or audiovisual content (Mukhopadhyay et al., 2024; Zhang et al., 2023; Tan et al., 2024)), have caused substantial societal harm, including misinformation, reputation damage, and financial losses. Real-world incidents and human psychology study alike have shown that the combination of audio and video significantly increases the deceptive power of deepfakes, with most harmful cases involving multimodal forgeries, underscoring the urgent need for MultiModal DeepFake Detection (MM-DFD)[1] to mitigate the growing threats.

Compared to deepfake detection relying on single modalities (*i.e.*, image (Chollet, 2017; Luo et al., 2021), audio (Radford et al., 2023; Jung et al., 2022), or video (Concas et al., 2024; Qiao et al., 2024)), MM-DFD reveals cross-modal inconsistencies and can effectively detect audiovisual deepfakes, with improved robustness and accuracy. However, MM-DFD receives relatively less attention due to two key gaps. First, there is a lack of a large-scale and comprehensive dataset that simulates realistic forgery techniques. Although several multimodal deepfake datasets have been introduced (Khalid et al., 2021; Cai et al., 2023; 2024; Xu et al., 2024), they suffer from three major limitations. **(1) Limited scale**: Except for the recently released AVDeepfake1M (Cai et al., 2024), which contains 860,039 fake samples, the scale of fake data in other datasets remains relatively small.

---

[1]In this work, "multimodal content" specifically refers to the content with audiovisual modality, distinguishing it from unimodal (video-only or audio-only) cases.

**(2) Low forgery diversity**: Most datasets are generated using only a few forgery methods (*e.g.*, AVDeepfake1M uses three while FakeAVCeleb uses four), making detectors prone to overfitting to specific forgery artifacts and failing to generalize to the wide variety of forgeries encountered in the wild. **(3) Low quality**: Existing datasets often contain visible unimodal artifacts or synchronization mismatches between modalities (Xu et al., 2024; Liu et al., 2025). These fidelity gaps between synthetic and real-world forgeries compromise the effectiveness of detectors trained on such data. In addition, there is an absence of a unified benchmark, preventing fair evaluation and comparison of detection methods.

In this work, we first propose **Mega-MMDF**, a large-scale multimodal deepfake dataset constructed through an automated deepfake generation framework. Compared to existing datasets, it has three key features: **(1) Large scale**: We generate 1.1 million forged audiovisual samples based on 0.1 million real samples, resulting in a total of 1.2 million instances—one of the largest MM-DFD datasets to date (to the best of our knowledge). **(2) Highest forgery diversity**: The dataset is constructed using 21 compositionally diverse forgery pipelines based on combinations of 28 forgery methods across 7 forgery types. **(3) Highest quality**: Our construction framework integrates a quality control mechanism to ensure high-quality audio, video, and audiovisual synchronization (see Table 1).

Moreover, following the establishment of DeepfakeBench (Yan et al., 2023) for visual deepfake detection, we argue that a unified benchmark is equally vital for MM-DFD. Unified benchmarks propel research through two fundamental mechanisms: (1) standardizing training/evaluation protocols across detection components, and (2) providing a common platform for method analysis and future investigations. However, intrinsic disparities between unimodal and multimodal data necessitate a dedicated MM-DFD benchmark, particularly in data preprocessing and model architectures. This critical infrastructure gap remains unaddressed in the current literature. Thus, we further develop **DeepfakeBench-MM**, the first unified benchmark for multimodal deepfake detection. It excels in three major aspects: **(1) Extensible modular-based codebase**: The benchmark codebase is designed with modularity and extensibility in mind, specifically supporting multimodal deepfake detection. **(2) Comprehensive evaluations**: We conduct comprehensive evaluations of 11 multimodal detectors on 5 multimodal datasets. **(3) Extensive analyses and novel findings**: Based on these evaluations, we provide in-depth analyses from multiple perspectives and uncover several novel insights, such as augmentation, finetuning, and generalization in MM-DFD.

Our key contributions are threefold: **(1)** We build a large-scale and diverse multimodal deepfake dataset **Mega-MMDF**, with 1.1 million high-fidelity fake audiovisual samples, constructed by 21 forgery pipelines based on 28 forgery methods of 7 forgery types. **(2)** We develop the first unified benchmark **DeepfakeBench-MM**, implemented with an extensible modular-based codebase. It supports 5 multimodal deepfake datasets and 11 multimodal deepfake detectors. **(3)** Based on DeepfakeBench-MM, we further provide **comprehensive evaluations and extensive analyses**, which reveal several **novel findings** to inspire more innovative research in this area. **In summary**, we believe that our dataset and benchmark provide a stepping stone for the advancement of multimodal deepfake detection research.

## 2 RELATED WORK

**Multimodal Deepfake Detectors** Beyond unimodal detectors focused on either audio deepfake detection (Radford et al., 2023; Jung et al., 2022) or visual deepfake detection (Chollet, 2017; Luo et al., 2021; Cao et al., 2022; Liu et al., 2021), multimodal deepfake detectors leverage both audio and visual modalities, emphasizing cross-modal consistency and feature fusion. To capture cross-modal consistency, MDS (Chugh et al., 2020) minimizes the Euclidean distance between audio and visual features for real samples and increases it for fake ones. AVTS (Sung et al., 2023) employs a contrastive loss during pretraining on real data, followed by a classification head trained on deepfake data. Furthermore, MRDF (Zou et al., 2024) introduces unimodal-specific regularizations and a multimodal classification head to handle cases where one or both modalities are manipulated. Another major direction is the cross-modal feature fusion. AVFF (Oorloff et al., 2024) adopts a structure inspired by MAE (He et al., 2022), incorporating a complementary masking strategy and cross-modal fusion to enhance representation learning. FRADE (Nie et al., 2024) introduces an AFI adapter to improve attention to high-frequency details and an ACI module to fuse audio and visual features. AVH (Smeu et al., 2025) leverages unimodal features extracted by a pretrained AV-HuBERT (Shi et al., 2022) model to give predictions after unsupervised or supervised training. Additionally, pretrained multimodal large language models, such as VideoLLaMA2 (Cheng et al., 2024) and Qwen2.5-Omni

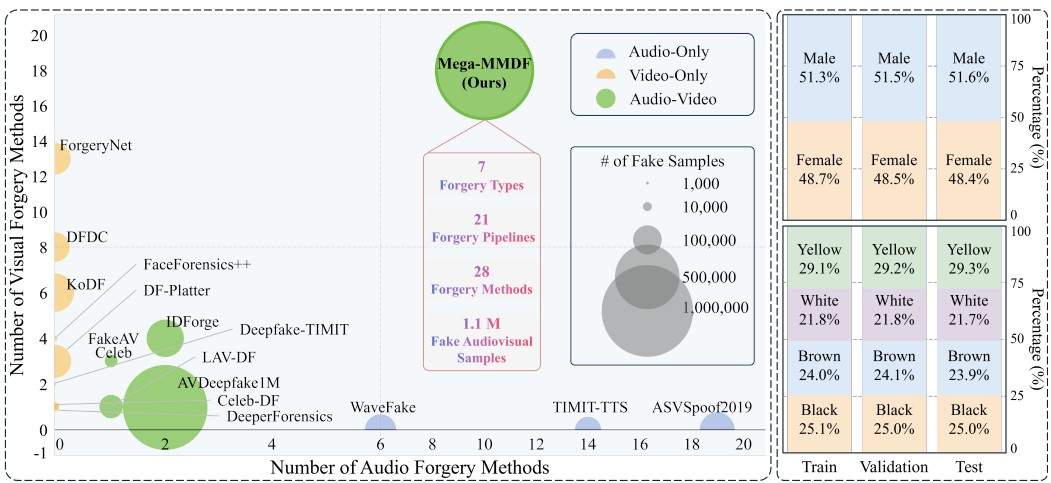

Figure 1: **Left**: Statistics of our **Mega-MMDF** dataset, demonstrating significant advantages in *scale* and *forgery diversity*. Note that only audiovisual samples in each dataset are counted. **Right**: Distributions of gender (top) and skin tone (bottom) in train/validation/test sets of Mega-MMDF.

(Xu et al., 2025), have demonstrated potential for multimodal deepfake detection in a zero-shot setting. Finally, several works (Cai et al., 2024; Thakral et al., 2025) explore ensemble methods, which combine the results of independently trained audio and visual models. These approaches serve as competitive baselines and are compared against multimodal detectors.

**Deepfake Datasets and Benchmarks**  As deepfake generation and detection techniques evolve, the demand for large-scale and diverse datasets continues to grow. Several surveys (Gong & Li, 2024; Pei et al., 2024; Altuncu et al., 2024) have provided comprehensive overviews of single-modality datasets, spanning image-, audio-, and video-based forgeries. However, the rise of multimodal audiovisual deepfakes—where both modalities are manipulated—has underscored the need for dedicated multimodal deepfake datasets. Notable examples include FakeAVCeleb (Khalid et al., 2021), IDForge (Xu et al., 2024), and ILLUSION (Thakral et al., 2025), which offer audiovisual forgeries. In addition, LAV-DF (Cai et al., 2023) and AVDeepfake1M (Cai et al., 2024) focus on temporal forgery localization, reflecting the increasing subtlety and complexity of deepfake techniques. However, existing multimodal datasets remain limited in scale and forgery diversity (see Table 5 and Figure 1 for comparisons). Moreover, there remains a notable gap in multimodal deepfake detection benchmarks that enable consistent evaluation across detectors and datasets. Although (Khalid et al., 2021; Thakral et al., 2025) propose several protocols, they lack unified training and testing settings and cover only a limited range of detectors and datasets, resulting in unfair comparison. For instance, FakeAVCeleb does not provide an official train/test split, leading subsequent studies to adopt inconsistent dataset partitions. To bridge this gap, we introduce DeepfakeBench-MM, alongside Mega-MMDF, a large-scale, diverse, and high-quality multimodal dataset. Together, they provide a unified platform for training and evaluating multimodal deepfake detectors.

## 3 MEGA-MMDF: THE MOST DIVERSE MILLION-SCALE MULTIMODAL DEEPFAKE DATASET

We aim to build a multimodal deepfake dataset named **Mega-MMDF**, which has three critical characteristics: **large scale**, **high forgery diversity**, and **high quality**. Thus, we develop an automated data construction framework, whose structure is detailed below.

### 3.1 REAL DATA COLLECTION

Our real-world audiovisual samples are drawn from three public datasets: Voxceleb2 (Chung et al., 2018), MEAD (Wang et al., 2020a), and CelebV-HQ (Zhu et al., 2022), with two key principles in mind: *diversity* and *balance*. **(1) Diversity**: The samples from these three datasets vary in resolution, speaker demographics, and recording environments, enabling us to capture a broad spectrum of real-world scenarios. **(2) Balance**: A well-balanced dataset is essential to avoid biased model performance. These datasets collectively provide a balanced representation of two critical demographic attributes: gender and skin tone. Specifically, we categorize the data into eight subgroups

defined by a combination of two genders (male and female) and four skin tones (white, black, yellow, and brown). Each subgroup contains 12,500 randomly selected samples, resulting in a balanced dataset of 100,000 real audiovisual samples with an average duration of 5.76 seconds. More statistics of our dataset can be found in Appendix A.4. Additionally, we adhere to the dataset licenses specified in Appendix A.5.

## 3.2 FORGERY PIPELINE

As shown in Figure 2, a **forgery pipeline** refers to *the complete process of constructing multimodal forged data from real data*, *which involves three forgery components*: *audio forgery*, *visual forgery*, and *audio-driven face reenactment*. We develop **21 compositionally diverse forgery pipelines** through systematic combinations of 10 audio forgery, 12 visual forgery, and 6 audio-video face reenactment methods, totaling 28 individual forgery methods.

**(1) Audio Forgery**: Following previous audio forgery works (Wang et al., 2020b; Wu et al., 2015), we consider three common types of audio forgeries: Text-to-Speech (TTS), Voice Conversion (VC), and Partial Fake (PF). Given a text input and reference audio, TTS generates fake audio adhering to the speaker's tone of the reference audio. VC converts one speaker's tone to another without altering the content of the speech. PF

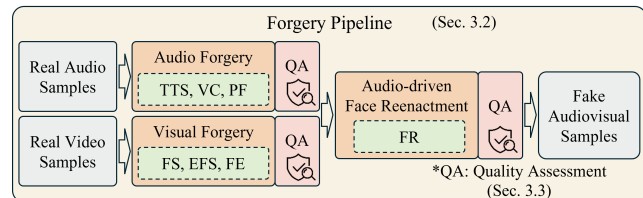

Figure 2: Illustration of the multimodal forgery pipeline, showing the steps of data construction.

modifies a specific part of the speech, such as deleting or changing a word, leveraging TTS and VC techniques. Specifically, we implement *5 TTS, 3 VC, and 2 PF methods*. **(2) Visual Forgery**: Following previous visual forgery works (Yan et al., 2024; 2023; Cai et al., 2024), we consider three common types of visual-only forgeries: Face Swapping (FS), Entire Face Synthesis (EFS), and Face Editing (FE). FS swaps the facial ID of the target video with the ID of the source video, while preserving all remaining contents (*e.g.*, expression, head pose, hair) of the target video. EFS generates an image of a non-existent face according to text prompts. FE edits the facial characteristics in an image according to text prompts, such as adding glasses or turning the black hair into white. Specifically, we implement *4 FS, 5 EFS, and 3 FE methods*. **(3) Audio-driven Face Reenactment**: After creating the unimodal forgeries, we ensure temporal and semantic coherence between modalities. For visual forgery methods that output static images, we employ audio-driven face reenactment (FR) to generate temporally consistent videos that align with the audio. Similarly, for video-based visual forgeries where the manipulated video does not naturally synchronize with the forged audio, we apply face reenactment to enforce cross-modal consistency, mimicking real-world deepfake scenarios. Specifically, we incorporate *6 FR methods*.

Finally, based on the forgery state (*real* or *fake*) of each modality, all samples are categorized into four groups: **RARV** (Real Audio Real Video), **RAFV** (Real Audio Fake Video), **FAFV** (Fake Audio Fake Video), and **FARV** (Fake Audio Real Video). Details of the 28 forgery methods and the 21 forgery pipelines are provided in Appendix A.1 and A.2. The illustrations of pipelines are provided in Figures 7, 8, 9, 10, 11, and 12.

## 3.3 QUALITY CONTROL MECHANISM DURING DEEPFAKE DATA CONSTRUCTION

To ensure the quality of the constructed forgery data, we design a **quality control mechanism** comprising three assessment modules: *audio quality assessment*, *visual quality assessment*, and *audio-video synchronization quality assessment*. As illustrated in Figure 2, each assessment module is integrated into the corresponding forgery component during the data construction process. The process is terminated if any assessment score falls below a predefined threshold. **(1) Audio Quality Assessment**: We focus on the audio quality metrics, including *audio fidelity* and *signal-to-noise ratio (SNR)*. **(a)** Audio fidelity measures the consistency of the fake audio content with the intended fake audio text, and we assess it using WhisperX (Bain et al., 2023), a speech-to-text (STT) model. **(b)** SNR quantifies the ratio of meaningful signal to background noise. We implement it by applying a robust SNR distance metric (Yuan et al., 2019). **(2) Visual Quality Assessment**: For the visual quality metric, we adopt the BRISQUE (Mittal et al., 2012) score, which uses scene statistics to quantify possible losses of "naturalness" in the frames due to distortions such as blur and blocking. The

BRISQUE metric is widely used in previous works (Narayan et al., 2023; Concas et al., 2024; Thakral et al., 2025; Wang et al., 2022) to assess visual quality due to its efficiency and strong correlation with human visual perception. The temporal LPIPS (Zhang et al., 2018) score is also leveraged to evaluate perceptual differences in the temporal dimension. **(3) Audio-Video Synchronization Quality Assessment**: The audio-video synchronization metrics we adopt include the Sync-C and Sync-D scores (Chung & Zisserman, 2017), which have been widely employed in previous face reenactment studies (Yang et al., 2023; Hegde et al., 2023; Lee et al., 2024) to assess the lip synchronization quality of manipulated audiovisual data.

### 3.4 Overall Quality

After the dataset construction process, we validate overall dataset quality through **(1) Objective Quantitative Metrics** using Fréchet Audio Distance (FAD) (Kilgour et al., 2019) and Fréchet Video Distance (FVD) (Unterthiner et al., 2018) to measure audio/video fidelity by quantifying real-forgery sample distribution gaps, and to evaluate synchronization using Sync-C and Sync-D calculated via a pretrained SyncNet (Chung & Zisserman, 2017) model. **(2) Human Studies** involving 60 volunteers from diverse backgrounds (*e.g.*, management, humanities, engineering), where a lower participant detection score indicates higher dataset fidelity (see Appendix A.3 for details).

Table 1 demonstrates our dataset's significant improvement in both objective metrics and human studies compared to FakeAVCeleb (Khalid et al., 2021), IDForge (Xu et al., 2024), and ILLUSION (Thakral et al., 2025) (whose FAD score is derived from its original manuscript, as the dataset remains unreleased), while excluding LAV-DF (Cai et al., 2023) and AVDeepfake1M (Cai et al., 2024) due to their focuses on localization tasks where only a small proportion of

Table 1: Overall quality assessments of different datasets. The best results are highlighted in **bold**.

| Dataset | Objective Metrics | | | | Human Study | |
|---|---|---|---|---|---|---|
| | FAD ↓ | FVD ↓ | Sync-D ↓ | Sync-C ↑ | ACC ↓ | AUC ↓ |
| FakeAVCeleb | 6.60 | 136.41 | 8.974 | 5.131 | 0.750 | 0.932 |
| IDForge | 4.30 | 287.67 | 11.056 | 3.525 | 0.475 | 0.873 |
| ILLUSION | 9.43 | - | - | - | - | - |
| **Ours** | **2.77** | **106.27** | **8.750** | **5.521** | **0.375** | **0.770** |

real samples are manipulated, making them inherently closer to real distribution. The result shows the effectiveness of our quality control mechanism throughout the forgery pipeline.

## 4 DeepfakeBench-MM: the first unified benchmark for multimodal deepfake detection

Built on our large-scale and highly diverse Mega-MMDF, we develop **DeepfakeBench-MM**, the first unified benchmark for multimodal deepfake detection. Supporting 5 multimodal deepfake datasets and 11 detectors, it provides the most comprehensive and rigorous evaluation framework that fills a critical gap in the field.

### 4.1 Supported Datasets and Detectors

**Datasets**  DeepfakeBench-MM currently supports 5 publicly available multimodal deepfake datasets: FakeAVCeleb (Khalid et al., 2021), LAV-DF (Cai et al., 2023), AVDeepfake1M (Cai et al., 2024), IDForge (Xu et al., 2024), and our Mega-MMDF. Although these datasets are well-structured, they exhibit significant heterogeneity in several aspects, such as storage format, file arrangements, and annotation standards. This poses challenges for fair evaluations and model development. To address this, we develop a standardized preprocessing pipeline and provide preprocessed features for each dataset to minimize redundant efforts. In addition, motivated by the leading silence issue reported in (Smeu et al., 2025), we skip the first two frames (80ms) during preprocessing to avoid potential leading silence shortcut. More details of preprocessing are provided in Appendix B.1.

**Multimodal Deepfake Detectors**  DeepfakeBench-MM currently implements 11 multimodal detectors, which span four categories: **(1) Baseline Models:** We implement a two-phase vanilla model consisting of a visual backbone, an audio backbone, and an MLP classifier. The backbones are first pretrained with a contrastive loss (*e.g.*, InfoNCE (Oord et al., 2018)) on a real audiovisual dataset, and then the MLP classifier is trained on fused features from a deepfake dataset. **(2) Regular Models:** These detectors are based on standard backbones such as CNNs or ViTs. We implement 7 widely used detectors, including AVTS (Sung et al., 2023), MRDF (Zou et al., 2024), AVFF (Oorloff et al., 2024), MDS (Chugh et al., 2020), FRADE (Nie et al., 2024), AVAD (Feng et al., 2023a), and AVH (Smeu et al., 2025). Notably, since AVFF is closed-source and AVTS, FRADE, and AVAD are only partially open-source, we reproduce these models following their original descriptions to integrate them into DeepfakeBench-MM to ensure fair comparison. **(3) Ensemble Models:** We implement an

Table 2: Performance (AUC) of multimodal deepfake detectors across multiple datasets. Definitions of the vanilla baseline and ensemble models are provided in Section 4.1. **Bold** denotes the best result, and underline denotes the second-best.

| Detector | FakeAVCeleb | IDForge | LAV-DF | AV-Deepfake1M | Mega-MMDF (Ours) |
|---|---|---|---|---|---|
| **Baseline and Ensemble Methods** (C3D-ResNet18 (Tran et al., 2017), SE-ResNet18 (Hu et al., 2018)) | | | | | |
| Baseline | 0.657 | 0.508 | 0.606 | 0.564 | 0.969 |
| Ensemble | 0.792 | 0.684 | 0.455 | 0.534 | 0.987 |
| **Regular Models** | | | | | |
| AVTS (Sung et al., 2023) | 0.678 | 0.598 | 0.496 | 0.511 | 0.847 |
| MRDF (Zou et al., 2024) | 0.663 | 0.517 | 0.606 | 0.530 | 0.970 |
| AVFF (Oorloff et al., 2024) | 0.414 | 0.501 | 0.568 | 0.486 | 0.858 |
| MDS (Chugh et al., 2020) | 0.606 | 0.544 | 0.601 | 0.525 | 0.806 |
| FRADE (Nie et al., 2024) | **0.868** | **0.771** | 0.668 | 0.507 | **0.994** |
| AVAD (Feng et al., 2023b) | 0.674 | 0.545 | 0.551 | 0.519 | 0.826 |
| AVH (Smeu et al., 2025) | 0.678 | 0.510 | **0.870** | **0.614** | 0.918 |
| **Pretrained Multimodal Large Language Models** | | | | | |
| Qwen2.5-Omni (Xu et al., 2025) | 0.605 | 0.571 | 0.650 | 0.570 | 0.568 |
| VideoLLaMA2 (Cheng et al., 2024) | 0.624 | 0.563 | 0.659 | 0.545 | 0.534 |

ensemble model that uses the same backbones as AVTS and the vanilla baseline. Predictions from the visual and audio detectors are averaged to obtain the final output. **(4) Pretrained Multimodal Large Language Models (MLLMs):** We include two pretrained MLLMs Qwen2.5-Omni (Xu et al., 2025) and VideoLLaMA2 (Cheng et al., 2024) as zero-shot detectors. Since large-scale, annotated audiovisual deepfake datasets suitable for MLLM finetuning are not yet available, we do not finetune these models but directly evaluate them in a zero-shot setting. Implementation details of these detectors are presented in Appendix B.2.

## 4.2 MAIN EVALUATIONS

Based on the 5 supported datasets, 11 multimodal detectors, and our benchmark codebase, we conduct comprehensive evaluations focusing on: **(1) intra-dataset performance**, **(2) cross-dataset generalization**, and **(3) cross-pipeline generalization**. For consistency with prior work (Yan et al., 2023; 2024), we report results using the widely adopted AUC metric.

**Intra-dataset Evaluation** All detectors are trained on the training set of our Mega-MMDF dataset, with the validation set used for model selection. The model attaining the highest performance on the validation set is subsequently evaluated on the test set. Mega-MMDF is adopted as the primary training set, instead of the previously widely used FakeAVCeleb, for three main reasons: **(1)** it is substantially larger and more diverse, encompassing 21 forgery pipelines that enable comprehensive evaluation; **(2)** it offers higher fidelity and incorporates more advanced deepfake generation methods (see Tables 1 and 6); and **(3)** it provides a more balanced fake-to-real ratio (11:1) compared to the extreme imbalance in FakeAVCeleb (approximately 42:1). Additional training and evaluation details are provided in Appendix B.2. As shown in Table 2, several detectors achieve strong performance, with MRDF (Zou et al., 2024) and FRADE (Nie et al., 2024) obtaining AUC scores of 0.970 and 0.994. Notably, FRADE sets the state-of-the-art result on our benchmark. Its success can be attributed to the joint modeling of audio–visual interactions within the backbone and its effective adaptation of forgery-relevant knowledge into pretrained ViT architectures.

**Cross-dataset Generalization** We next examine how detectors generalize across datasets with varying forgery types and distributions. We evaluate the detectors on the test sets of FakeAVCeleb, IDForge, AVDeepfake1M and LAV-DF. LAV-DF and AVDeepfake1M are localization datasets containing partially manipulated samples; they are included to evaluate whether detectors can effectively identify such partial forgeries. As shown in Table 2, except for AVH, detector performance drops substantially on partial-fake datasets like LAV-DF and AVDeepfake1M. We attribute this gap to distributional and structural differences in forgeries across datasets. In LAV-DF and AVDeepfake1M, manipulations are confined to short temporal segments or localized visual regions, which may not activate strong **modality-specific** artifacts learned from fully forged samples in Mega-MMDF. By contrast, AVH benefits from pretrained lip-reading checkpoints, enabling it to exploit **cross-modal** consistency and achieve superior performance in detecting partial manipulations. Moreover, we note

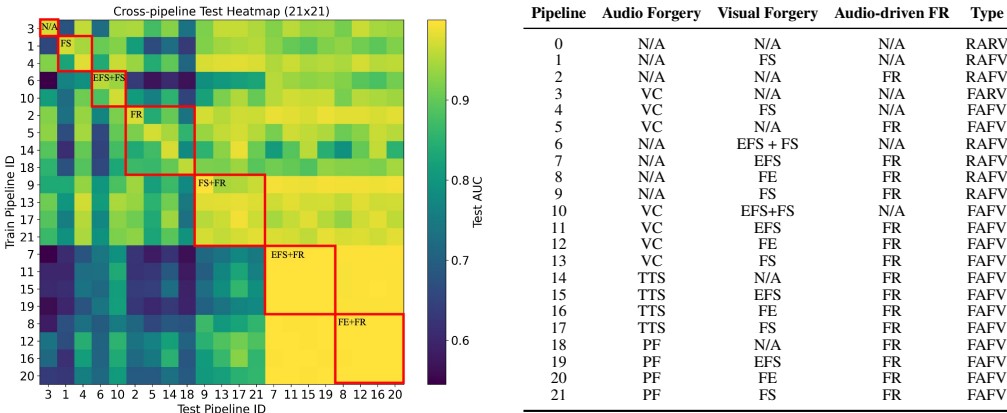

Figure 3: Cross-pipeline One-Versus-All (OvA) protocol. **Left:** The cross-pipeline performance heatmap, where red bounding boxes highlight generalization of pipelines employing the same visual forgeries. **Right:** Detailed illustrations of 22 pipelines in our Mega-MMDF dataset.

that pretrained large models, when evaluated in a zero-shot setting, obtain the lowest AUC scores on Mega-MMDF, highlighting both the difficulty and the novelty of our dataset.

**Cross-pipeline Generalization** We train the baseline multimodal detector on each forgery pipeline (together with real data) and evaluate its performance across all the pipelines. We emphasize the concept of *forgery pipelines* because multimodal deepfake data typically involve combinations of audio forgery, visual forgery, and audio–video synchronization, rather than a single unimodal manipulation. With 21 distinct pipelines, Mega-MMDF serves as an ideal testbed for studying cross-pipeline generalization. Given the strong reliance on visual cues (see Analysis 4), we group forgery pipelines according to shared visual artifacts. This grouping also facilitates the analysis of *stacked artifacts* (*e.g.*, EFS+FR), which combine multiple manipulation types. As shown in Figure 3, while the EFS+FR pipeline exhibits strong bidirectional generalization with FE+FR, likely due to shared diffusion artifacts between EFS and FE, the majority of pipeline groups display pronounced asymmetry in cross-pipeline generalization, evident from the non-symmetric patterns in the AUC matrix. This asymmetry suggests that the ability of a model trained on one pipeline to generalize to another is not necessarily reciprocal, pointing to potential modality dominance or artifact-specific dependencies. This phenomenon is further examined in detail in Analysis 2 (see below).

## 4.3 ANALYSIS AND FINDINGS

**Analysis 1: How do different detectors perform within and across datasets?** From the intra- and cross-dataset evaluations in Table 2, we observe that although detectors generally achieve high AUCs within Mega-MMDF (up to 0.994), their performance drops markedly when applied to external datasets. This contrast highlights both the strength of in-domain adaptation and the challenges of cross-domain generalization, motivating a deeper examination of detector design and fusion strategies. Two notable observations emerge:

(1) **Baseline *vs*. Others.** Our baseline achieves an intra-dataset AUC of 0.969, outperforming several more sophisticated methods. This suggests that, under identical training protocols, the marginal gains offered by advanced detectors over a simple baseline are limited. In cross-dataset evaluations, regular models also fail to consistently surpass the baseline, indicating that our baseline serves as a strong reference point for future multimodal deepfake detection research.

(2) **Ensemble *vs*. Others.** Our ensemble approach, built from the same backbones as AVTS and the baseline, achieves superior results compared to both, demonstrating that ensemble models can serve as another competitive baseline. The timing and method of fusing modality-specific features play a critical role; effective fusion allows multimodal detectors to surpass simple averaging of unimodal predictions and better exploit complementary information across modalities.

**Analysis 2: What drives asymmetric generalization across pipelines?** From the cross-pipeline evaluation in Figure 3, we observe that while certain forgery types generalize symmetrically, many exhibit pronounced non-reciprocal behavior, particularly in stacked forgeries combining multiple artifacts. To examine the source of these asymmetries, we analyze the relative influence of different artifacts within such combinations:

(1) **Effect of EFS *vs.* FS.** In the cross-pipeline heatmap, EFS+FS generalizes much better to other EFS-related pipelines (*e.g.*, EFS+FR) than FS-only pipelines, suggesting that EFS+FS retains stronger EFS-related features. The t-SNE visualization in Figure 4-right supports this, showing clearer separation for EFS+FR than FS, when trained on EFS+FS.

(2) **Effect of FS *vs.* FR.** Although less apparent in the heatmap, the t-SNE plots in Figure 4 reveal that training on EFS+FR yields little separation for FR, whereas training on EFS+FS produces noticeable separation for FS. This implies that FS contributes more than FR when combined with EFS and that EFS remains dominant in combinations such as EFS+FR.

Taken together, these findings indicate that in stacked forgeries, artifacts contribute unequally, following the hierarchy EFS > FS > FR. The dominance of EFS in certain combinations reflects an **artifact bias phenomenon**, where specific artifacts disproportionately influence the learned representation. This bias can skew cross-pipeline generalization and highlights *the need for future work on representation balancing to ensure fairer exploitation of multi-artifact information*.

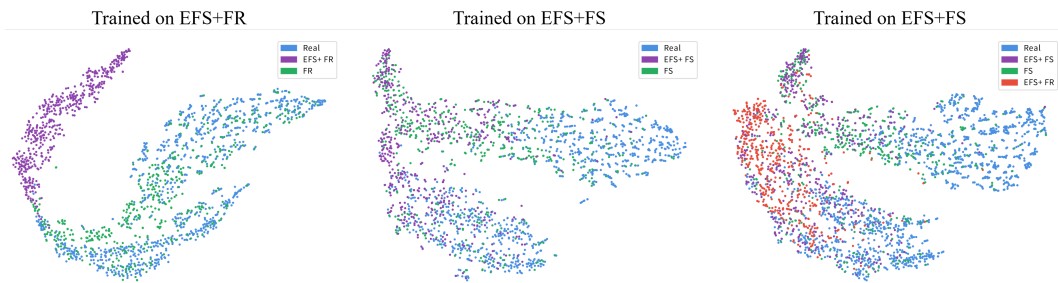

Figure 4: t-SNE visualizations of the baseline model on the cross-pipeline setting.

**Analysis 3: Does finetuning pretrained backbones benefit multimodal deepfake detection?**
Several multimodal deepfake detectors, such as AVFF and AVTS, adopt a two-phase paradigm. Backbones are first pretrained on large-scale real audio-video pairs to learn cross-modal synchronization, and then trained on deepfake datasets for classification. In the second stage, these backbones are typically frozen to preserve synchronized features (Sung et al., 2023). We hypothesize that this freezing strategy may be *overly conservative*, potentially limiting adaptation to forgery-specific artifacts. To test this, we finetune the backbones of AVTS, AVFF, and our vanilla baseline. As shown in Table 3, finetuning yields *consistent improvements across datasets*. These results suggest that **finetuning allows models to adapt backbone representations to artifact-specific cues essential for detection** without compromising the pre-learned cross-modal features. *This finding challenges the prevailing assumption that frozen backbones are necessary in the second stage and indicates that controlled finetuning can be a practical design choice for stronger multimodal forgery detectors.*

Table 3: Results of finetuning the second-stage backbones of two-phase models and applying random modality masking as data augmentation. For each $a/b$ pair: $a$ is the AUC of the detector without finetuning or masking, while $b$ is that of the detector trained with finetuning or masking. The best result in each pair is highlighted in **bold**.

|  | FakeAVCeleb | IDForge | LAV-DF | AV-Deepfake1M | Mega-MMDF (Ours) |
|---|---|---|---|---|---|
| Baseline (w/ f.t.) | 0.657 / **0.665** | 0.508 / **0.546** | 0.606 / **0.623** | 0.564 / **0.565** | 0.969 / **0.983** |
| AVTS (w/ f.t.) | 0.678 / **0.784** | 0.598 / **0.688** | 0.496 / **0.506** | 0.511 / 0.511 | 0.847 / **0.983** |
| AVFF (w/ f.t.) | 0.414 / **0.848** | 0.501 / **0.662** | 0.568 / **0.654** | 0.486 / **0.588** | 0.858 / **0.997** |
| AVTS (w/ aug) | 0.678 / **0.689** | 0.598 / **0.612** | 0.496 / **0.515** | 0.511 / **0.522** | 0.847 / **0.863** |
| MRDF (w/ aug) | 0.663 / **0.697** | 0.517 / **0.599** | 0.606 / **0.696** | 0.530 / **0.544** | 0.970 / **0.974** |

**Analysis 4: Do multimodal deepfake detectors perform equally well across single modalities?**
To examine performance balance across modalities, we evaluate each detector on single-modality forgeries by masking either the audio or video stream and using the ground-truth label of the unmasked modality. The results are reported in Table 4. Models designed to inherently require both modalities, such as AVAD, MDS, and AVH, are excluded from this experiment. The evaluation reveals a **consistent bias toward the visual modality**: detectors generally achieve higher AUC with video-only input compared to audio-only input. This indicates a notable performance imbalance when multimodal detectors operate in a single-modality scenario. A plausible explanation is the well-documented **modality bias** (Huang et al., 2022; Wu et al., 2024; Chaudhuri et al., 2025), in which

multimodal models fail to leverage both modalities in a balanced manner. Since most detectors adopt a late-fusion architecture, the two modalities often act in competition rather than complementarity. This architecture can lead to predictive cues from one modality to become entangled with noise features from the other, suppressing informative signals (Chaudhuri et al., 2025). While such modality bias has been extensively discussed in multimodal representation learning, it remains largely unaddressed in current multimodal deepfake detection frameworks, highlighting a critical design limitation that suggests *future detectors should incorporate strategies, such as bias-aware training or adaptive fusion, to mitigate modality dominance and fully exploit the strengths of each modality*.

Table 4: Performance of detectors on unimodal and multimodal input.

| Input Type | Baseline | AVTS | AVTS (w/ aug) | MRDF | MRDF (w/ aug) | AVFF | FRADE |
|---|---|---|---|---|---|---|---|
| Video | 0.517 | 0.838 | 0.842 | 0.928 | 0.953 | 0.816 | 0.995 |
| Audio | 0.516 | 0.638 | 0.693 | 0.700 | 0.792 | 0.766 | 0.897 |
| Multimodal | 0.969 | 0.847 | 0.863 | 0.970 | 0.974 | 0.858 | 0.994 |

**Analysis 5: Can modality masking mitigate modality bias?** Motivated by the range of potential strategies outlined in the previous analysis to address modality bias, we make a preliminary attempt by randomly masking the dominant modality during training. The aim is to encourage the model to better exploit information from the weaker modality and, in turn, improve overall detection performance. This approach can be regarded as a bias-mitigation form of data augmentation, which is rarely explored in multimodal forgery detection. Given that models trained on Mega-MMDF generally exhibit a strong preference for the visual modality, we randomly mask the video stream during training and supervise using audio-modality labels. We evaluate this strategy on AVTS and MRDF, representing one-phase and two-phase architectures, respectively. As shown in Table 3, both intra-dataset performance and cross-dataset generalization improve. Furthermore, Table 4 demonstrates more balanced results, suggesting that **modality masking is a promising augmentation strategy for reducing modality dominance**. While these findings offer encouraging evidence that targeted suppression of the dominant modality can promote more balanced cross-modal learning, their generalizability across datasets, architectures, and bias patterns remains to be determined. *Future work should examine modality bias from a theoretical perspective and explore principled bias-mitigation techniques, such as adaptive fusion or bias-aware optimization, to achieve complementary use of all modalities in multimodal deepfake detection.*

### Take-away Messages and Future Directions

(1) **Regarding Model:** Current multimodal detectors only show marginal gains over the baselines and rely more on visual artifacts. This calls for new architectures and fusion strategies that mitigate modality bias and improve overall performance.

(2) **Regarding Training:** finetuning the second-stage backbones improves performance, while randomly masking the well-learned modality during training yields more balanced results. Future work should explore broader training strategies to further equalize modality contributions.

(3) **Regarding Data:** Detection is largely driven by dominant EFS and FE artifacts in the visual modality, limiting cross-pipeline generalization. Mitigating this artifact bias is key to learning richer and more diverse forgery cues.

## 5 CONCLUSION

In this work, we first introduce **Mega-MMDF**, a million-scale, diverse, and high-quality dataset designed to advance research in multimodal deepfake detection. Building upon it, we develop **DeepfakeBench-MM**, the first unified benchmark that standardizes preprocessing, training, and evaluation protocols for multimodal deepfake detection. Supporting 5 multimodal deepfake datasets and 11 representative detectors within an extensible codebase, DeepfakeBench-MM enables fair, reproducible comparisons and facilitates the development of new methods. Leveraging these infrastructures, we conduct extensive experiments and uncover several novel insights into the strengths and limitations of current methods. We believe that the release of Mega-MMDF and DeepfakeBench-MM, together with the reported findings, will provide a solid foundation for future research.

**Summary of Appendix** Due to space limitation, additional details are presented in **Appendix**. **(1) Dataset**: Details of forgery methods and pipelines are elaborated in Appendices A.1 and A.2. Human study and dataset statistics are provided in Appendices A.3 and A.4. The licenses of utilized datasets are listed in Appendix A.5. **(2) Benchmark**: Details of benchmark codebase and detector settings are presented in Appendices B.1, B.2, and B.3. More analyses are provided in Appendix B.4. **(3) LLM usage**: The use of LLMs is declared in Appendix C.

## 6 ETHICS STATEMENT

Our work aims to promote trustworthy multimodal deepfake detection, thereby contributing to positive societal impacts by helping mitigate risks associated with the misuse of deepfake technologies. All datasets used in this work strictly adhere to their respective licenses, which we detail in Appendix A.5. Although our dataset is specifically designed to advance research in deepfake detection, we acknowledge the dual-use concern that its quality could inadvertently benefit deepfake generation. To mitigate this risk, we will enforce controlled distribution of the dataset and require recipients to sign a research-use agreement explicitly prohibiting its use for deepfake creation. Finally, the human study conducted to evaluate the realism of generated videos in our work was reviewed and approved by an IRB.

## 7 REPRODUCIBILITY STATEMENT

Our benchmark is designed to facilitate the reproduction of existing multimodal deepfake detection methods, which we believe is much needed in the field. DeepfakeBench-MM codebase is released at https://github.com/AnonymousDeepfakeBench-MM/DeepfakeBench-MM for anonymous review purpose. Moreover, research-only access to our Mega-MMDF dataset will be released after the review period. To further support reproducibility, we provide detailed descriptions of dataset preprocessing in Appendix B.1, and specify the implementation and training settings of all detectors in Appendix B.2.

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

# Appendix

# A DATASET

Table 5: Comparison of publicly available deepfake datasets. TTS: Text To Speech. VC: Voice Conversion. PF: Partial Fake. FS: Face Swapping. EFS: Entire Face Synthesis. FE: Face Editing. FR: Face Reenactment.

| Dataset | Year | Modality | Real Samples | Fake Samples | #Forgery Methods | TTS | VC | PF | FS | EFS | FE | FR | Stacked Forgery | Quality Control |
|---|---|---|---|---|---|---|---|---|---|---|---|---|---|---|
| ASVSpoof2019 (Wang et al., 2020b) | 2019 | A | 16,492 | 148,148 | 19 | 11 | 8 | - | - | - | - | - | ✔ | ✗ |
| WaveFake (Frank & Schönherr, 2021) | 2021 | A | 0 | 117,985 | 6 | 6 | - | - | - | - | - | - | ✗ | ✗ |
| TIMIT-TTS (Salvi et al., 2023) | 2022 | A | 430 | 80,000 | 14 | 14 | - | - | - | - | - | - | ✗ | ✗ |
| Deepfake-TIMIT (Korshunov & Marcel, 2018) | 2018 | V | 320 | 640 | 2 | - | - | - | 2 | - | - | - | ✗ | ✗ |
| FaceForensics++ (Rossler et al., 2019) | 2019 | V | 1,000 | 4,000 | 4 | - | - | - | 2 | - | - | 2 | ✗ | ✗ |
| Celeb-DF (Li et al., 2020) | 2020 | V | 590 | 5,639 | 1 | - | - | - | 1 | - | - | - | ✗ | ✗ |
| DFDC (Dolhansky et al., 2020) | 2020 | V | 23,654 | 104,500 | 8 | - | - | - | 5 | 2 | - | 1 | ✗ | ✗ |
| DeeperForensics (Jiang et al., 2020) | 2020 | V | 50,000 | 10,000 | 1 | - | - | - | 1 | - | - | - | ✗ | ✗ |
| KoDF (Kwon et al., 2021) | 2021 | V | 62,166 | 175,776 | 6 | - | - | - | 3 | - | - | 3 | ✗ | ✗ |
| ForgeryNet (He et al., 2021) | 2021 | V | 99,630 | 121,617 | 13 | - | - | - | 5 | - | 5 | 3 | ✔ | ✗ |
| DF-Platter (Narayan et al., 2023) | 2023 | V | 133,260 | 132,496 | 3 | - | - | - | 3 | - | - | - | ✗ | ✗ |
| FakeAVCeleb (Khalid et al., 2021) | 2021 | AV | 500 | 19,500 | 4 | - | 1 | - | 2 | - | - | 1 | ✔ | ✗ |
| LAV-DF (Cai et al., 2023) | 2022 | AV | 36,431 | 99,873 | 2 | - | - | 1 | - | - | - | 1 | ✔ | ✗ |
| IDForge (Xu et al., 2024) | 2024 | AV | 79,827 | 169,311 | 6 | - | 2 | - | 3 | - | - | 1 | ✔ | ✗ |
| AVDeepfake1M (Cai et al., 2024) | 2024 | AV | 286,721 | 860,039 | 3 | - | - | 2 | - | - | - | 1 | ✗ | ✗ |
| **Mega-MMDF (Ours)** | **2025** | **AV** | **100,000** | **1,100,000** | **28** | **5** | **3** | **2** | **4** | **5** | **3** | **6** | **✔** | **✔** |

## A.1 FORGERY METHOD

Information on 28 forgery methods 28 forgery methods across 7 forgery types is summarized in Table 6. Our focus is primarily on recent deepfake generation techniques that are commonly employed in real-world scenarios. In addition to methods published in academic venues, we include a diverse set of approaches sourced from the open-source community and industry. This comprehensive selection reflects the breadth and evolving nature of the current deepfake generation landscape.

Table 6: Implemented forgery methods of our Mega-MMDF dataset. We use the following methods to create multimodal deepfake samples.

| Modality | Forgery Type | Forgery Method | Year | Venue | Data Format |
|---|---|---|---|---|---|
| Audio | TTS (Text-To-Speech) | YourTTS (Casanova et al., 2022) | 2022 | ICML 2022 | Audio |
| | | SpeechT5 (Ao et al., 2022) | 2022 | ACL 2022 | Audio |
| | | OpenVoice V1 (Qin et al., 2023) | 2023 | Myshell-AI GitHub Star 30.5K | Audio |
| | | XTTS (Casanova et al., 2024) | 2024 | Interspeech 2024 | Audio |
| | | Vevo (Zhang et al., 2025) | 2025 | ICLR 2025 | Audio |
| | VC (Voice Conversion) | FreeVC (Li et al., 2023) | 2023 | ICASSP 2023 | Audio |
| | | Diff-HierVC (Choi et al., 2023) | 2023 | Interspeech 2023 | Audio |
| | | Vevo (Zhang et al., 2025) | 2024 | ICLR 2025 | Audio |
| | PF (Partial Fake) | FluentSpeech (Jiang et al., 2023) | 2023 | ACL 2023 | Audio |
| | | VoiceCraft (Peng et al., 2024) | 2024 | ACL 2024 | Audio |
| Visual | FS (Face Swapping) | Uniface (Xu et al., 2022) | 2022 | ECCV 2022 | Video |
| | | BlendFace (Shiohara et al., 2023) | 2023 | ICCV 2023 | Video |
| | | CSCS (Huang et al., 2024) | 2024 | ACM TOG 2024 | Video |
| | | InSwapper (Wang, 2023) | 2024 | GitHub Star 611 | Video |
| | EFS (Entire Face Synthesis) | ThisPersonDoesNotExist (Wang, 2019) | 2019 | Daily Visits 20K | Image |
| | | Stable-Diffusion-XL-Base (Podell et al., 2024) | 2024 | Hugging Face 2.46M | Image |
| | | RealVisXL_V3.0 (SG161222, 2025a) | 2024 | Hugging Face 47.1K | Image |
| | | Realistic_Vision_V4.0_noVAE (SG161222, 2025c) | 2024 | Hugging Face 30.6K | Image |
| | | RealVisXL_V5.0 (SG161222, 2025b) | 2024 | Hugging Face 70.1K | Image |
| | FE (Face Editing) | IP-Adapter (Ye et al., 2023) | 2023 | GitHub Star 5.9K | Image |
| | | PhotoMaker (Li et al., 2024b) | 2024 | CVPR 2024 | Image |
| | | PuLID (Guo et al., 2024) | 2024 | NeurIPS 2024 | Image |
| Audio-Video | FR (Face Reenactment) | Wav2Lip (Prajwal et al., 2020) | 2020 | ACMMM 2020 | Video |
| | | SadTalker (Zhang et al., 2023) | 2023 | CVPR 2023 | Video |
| | | MuseTalk (Zhang et al., 2024) | 2024 | GitHub Star 4K | Video |
| | | Diff2Lip (Mukhopadhyay et al., 2024) | 2024 | WACV 2024 | Video |
| | | EDTalk (Tan et al., 2024) | 2024 | ECCV 2024 | Video |
| | | LatentSync (Li et al., 2024a) | 2024 | GitHub Star 3.7K | Video |

## A.2 PIPELINE ILLUSTRATION

We design a total of 22 pipelines, including 1 real pipeline and 21 forgery pipelines, covering the four combinations of real and fake modalities: RARV (Real Audio, Real Video), RAFV (Real Audio, Fake Video), FARV (Fake Audio, Real Video), and FAFV (Fake Audio, Fake Video). For each pipeline, we first sample a pair of subjects and guide them through three stages: audio forgery, visual forgery, and audio-driven face reenactment. Note that at each stage, a sample may undergo no manipulation,

depending on the specific pipeline design. The detailed construction of all pipelines is provided on the right side of Figure 3. The illustrations of pipelines are also provided in Figures 7, 8, 9, 10, 11, and 12.

Importantly, quality control is integrated into every pipeline, as described in Section 3.3, to ensure the high fidelity and consistency of the generated fake data. Additionally, we show sample videos from Mega-MMDF in Figures 13, 14, 15, 16.

## A.3 HUMAN STUDY DETAILS

Participants are divided into 15 groups of 4 individuals each. Our IRB-approved human study consists of three experiments. Among the 15 groups, 10 participate in Experiment 1, 3 in Experiment 2, and 2 in Experiment 3. In **Experiment 1**, each group receives 12 short video clips covering four content types: RARV, FARV, RAFV, and FAFV. For each clip, participants determine whether the audio and video are real or fake and rate their confidence in the response. While Experiment 1 involves binary judgments, Experiments 2 and 3 are ranking tasks. In **Experiment 2**, each group views 8 video sets, each containing 3 fake videos from different datasets. After being informed of the forgery type, participants rank the videos in each set by perceived realism (from most to least realistic). In **Experiment 3**, each group is shown 6 video sets, again with 3 fake videos per set. Unlike Experiment 2, participants are not informed of the forgery type and must rank the videos based solely on perceived realism. Upon obtaining the data, we process it in the following steps:

**Judgment Task (Experiment 1):** We convert participants' answers and their confidence scores into probabilities. Let the confidence score be $c_i \in \{1, \ldots, 5\}$. The probability $\hat{p}_i$ that a video is fake is computed as:

$$\hat{p}_i = 0.5 + 0.1 \, c_i \, (2 \, \mathbb{I}[\text{fake}] - 1), \tag{1}$$

where $\mathbb{I}[\text{fake}]$ equals 1 if the participant labels the video as "fake," and 0 otherwise. We use the resulting probabilities to compute **Accuracy** and **AUC**, which reflect human performance in identifying deepfakes.

**Ranking Tasks (Experiments 2 and 3):** We calculate a composite score $S_j$ to evaluate the relative ranking of dataset $j$. Assume $n$ participants rank $m$ alternatives from most realistic to most fake. We assign a position-dependent weight $w_k$ to each rank:

$$w_k = m - k + 1, \qquad k = 1, \ldots, m. \tag{2}$$

Let $f_{jk}$ denote the number of times dataset $j$ is ranked in position $k$. The composite score $S_j$ is calculated as:

$$S_j = \frac{1}{n} \sum_{k=1}^{m} f_{jk} \cdot w_k. \tag{3}$$

We define **Score@2** and **Score@3** as the average values of $S_j$ obtained from Experiments 2 and 3, respectively.

As shown in Table 7, which presents the complete results of our human study, our dataset achieves the lowest Accuracy and AUC in Experiment 1, indicating that it is the most difficult for participants to judge. Furthermore, it achieves the highest ranking scores in both Experiment 2 and Experiment 3, suggesting that our dataset contains the most realistic forgeries from a human perceptual standpoint.

Table 7: Full results of quality assessment.

| Dataset | FVD↓ | FAD↓ | Sync-D↓ | Sync-C↑ | Human Study | | | |
|---|---|---|---|---|---|---|---|---|
| | | | | | ACC↓ | AUC↓ | Score@2↑ | Score@3↑ |
| FakeAVCeleb | 136.41 | 6.60 | 8.974 | 5.131 | 0.750 | 0.932 | 1.979 | 2.000 |
| IDForge | 287.67 | 4.30 | 11.056 | 3.525 | 0.475 | 0.873 | 1.825 | 1.813 |
| ILLUSION | - | 9.43 | - | - | - | - | - | - |
| **Mega-MMDF (Ours)** | **106.27** | **2.77** | **8.750** | **5.521** | **0.375** | **0.770** | **2.196** | **2.188** |

## A.4 DATASET STATISTICS

**Scale, Duration, and Storage** The complete Mega-MMDF has 1.2 million audio-visual samples (100,000 real and 1.1 million fake) of 1,723 hours in total length with an average duration of 5.17

seconds per sample. The data is saved in `MP4` format, and the total storage is 930 GB, with 73.43% of the samples being smaller than 1 MB.

**Train/Validation/Test Split** To standardize training and evaluation in our benchmark, we partition the entire dataset into train, validation, and test sets using an approximate 6:2:2 ratio. To maintain balance across gender and skin tone distributions, we first divide the dataset into 8 subgroups (2 genders × 4 skin tones) following the real data partition (see Section 3.1), then perform a 6:2:2 random split within each subgroup before merging corresponding subsets to form the final train/validation/test sets, with Figure 1 demonstrating well-balanced distributions across all partitions.

**Scale Statistics** We ultimately obtain a large-scale dataset comprising 1.2 million audio-visual samples, including 0.1 million real and 1.1 million fake ones. To ensure consistency and foster standardized training, we partition the dataset into training, validation, and test sets using a 6:2:2 ratio. Rather than applying a random split, we adopt the stratified sampling strategy from the quality control stage, ensuring each subgroup is divided using the same ratio. This maintains a balanced distribution of gender and skin tone across all splits, enhancing fairness in model training and evaluation. In terms of duration, the dataset spans a total of 6,203,873 seconds of content, with an average duration of 5.17 seconds per clip. Furthermore, the dataset is storage-efficient, with 73.43% of the clips being smaller than 1MB. These characteristics make the dataset both comprehensive and practical for large-scale deployment.

## A.5 LICENSES OF UTILIZED DATASETS

In this work, we incorporate four existing deepfake datasets for research purposes: FakeAVCeleb (Khalid et al., 2021), IDForge (Xu et al., 2024), AVDeepfake1M (Cai et al., 2024), and LAV-DF (Cai et al., 2023). Additionally, during the generation of Mega-MMDF, we use VoxCeleb2 (Chung et al., 2018), MEAD (Wang et al., 2020a), and CelebV-HQ (Zhu et al., 2022).

Below, we provide a brief overview of each dataset, including licensing information.

**FakeAVCeleb** FakeAVCeleb is one of the first works to propose a multimodal deepfake dataset aimed at addressing the growing threat of impersonation attacks using both synthetic video and audio. The dataset is constructed from real YouTube videos of celebrities spanning four ethnic backgrounds, enhancing both realism and diversity. It is distributed under the CC BY 4.0 license.

**IDForge** IDForge is an identity-driven multimedia forgery dataset. It contains 249,138 manipulated video shots generated from 324 wild videos of 54 celebrities. In addition, IDForge includes 214,438 real video shots as a reference set, enabling identity-aware forgery detection. Access is controlled, and users must apply via email for access. The license for the dataset is not publicly disclosed.

**AVDeepfake1M** AVDeepfake1M is a large-scale deepfake localization dataset designed to advance the detection and localization of subtle audio-visual manipulations. It includes around 860,000 fake videos and audio-visual manipulations across more than 2,000 subjects. It is distributed under the EULA (End-User License Agreement).

**LAV-DF** LAV-DF (Localized Audio-Visual DeepFake) is a localization deepfake dataset designed to support the detection and localization of subtle, content-driven manipulations in audio and video. LAV-DF focuses on temporally localized forgeries that can significantly alter a video's meaning—such as changing its sentiment polarity. It is distributed under the CC BY-NC 4.0 license.

**VoxCeleb2** VoxCeleb2 is a large-scale audio-visual speaker recognition dataset collected from open-source media. It contains 1,128,246 utterances from 6,112 celebrities, collected from YouTube videos. It is distributed under the CC BY-SA 4.0 license.

**MEAD** MEAD (Multi-view Emotional Audio-visual Dataset) is a talking-face video corpus featuring 60 actors and actresses talking with 8 different emotions at 3 different intensity levels. This dataset amounts to about 281,400 video clips at 1920×1080 resolution. There are no specific restrictions on usage.

**CelebV-HQ**   CelebV-HQ (High-Quality Celebrity Video Dataset) contains 35,666 video clips with the resolution of 512x512, involving 15,653 identities. This dataset is distributed under a non-commercial, research-only license that prohibits commercial use or redistribution for profit.

# B   BENCHMARK

Table 8: Comparison of previous evaluation protocols and our unified DeepfakeBench-MM. MM means MultiModal.

| | Year | Task | #MM Methods | #MM Datasets | Cross-MMDataset Eval | Cross-Pipeline Eval | Official Train/(Val)/Test Split | Code Access |
|---|---|---|---|---|---|---|---|---|
| FakeAVCeleb | 2021 | Detection | 2 | 1 | ✗ | ✗ | ✗ | ✔ |
| AVDeepfake1M | 2024 | Localization | 3 | 2 | ✔ | ✗ | ✔ | ✔ |
| ILLUSION | 2025 | Detection | 3 | 1 | ✗ | ✗ | ✗ | ✗ |
| **DeepfakeBench-MM (Ours)** | **2025** | **Detection** | **11** | **5** | ✔ | ✔ | ✔ | ✔ |

## B.1   DETAILS OF DATA PREPROCESSING

This section describes the data preprocessing utilities and the overall workflow. The preprocessing utilities offer various functions including separation of audio and video streams, audio resampling, video frame rate adjustment, face alignment and cropping, as well as audio segmentation. Building upon these utilities, the preprocessing workflow handles data preparation, metadata parsing, and processing tasks. To accelerate the workflow, parallel computing support is also integrated.

**Overall Workflow**   The preprocessing workflow begins by loading a configuration file and parsing metadata. The configuration specifies parameters such as the root directory of the original dataset, the output directory, and various preprocessing settings, including video frame rate, audio sampling rate, segmentation length, and the number of clips to extract per sample. Based on the specified data path, the metadata is parsed to retrieve information such as file paths, forgery methods, labels, and other relevant attributes. Each sample then undergoes 1) audio-video stream separation, audio resampling, and video frame rate adjustment to ensure consistent modality parameters, 2) face detection, alignment, and cropping are performed until the desired number of consecutive clips is obtained or the end of the video is reached. Finally, the preprocessed clips are saved either in MP4 format for reduced storage cost and ease of visualization, or in NPZ format for efficient data loading during training and evaluation. The details of each preprocessing step are described below.

**Audio-Video Separation and Adjustment**   In many publicly available datasets, audio and video are stored together in a single file, which complicates the process of loading audio independently. To address this issue, we utilize FFMPEG to separate the audio and video streams, producing a video-only MP4 file and an audio-only WAV file. During this process, we also adjust the video frame rate and resample audio. Based on a review of common settings in open-source detection methods, we adopt a standardized frame rate of 25 FPS and an audio sampling rate of 16 kHz.

**Face Detection and Alignment**   Inspired by DeepfakeBench (Yan et al., 2023) and AV-HuBERT (Shi et al., 2022), we perform face detection, alignment, and cropping on a frame-by-frame basis using Dlib (Sagonas et al., 2016), which provides efficient face detection algorithms and pretrained shape predictors. Once a face is detected in a frame, facial landmarks are extracted and key points around the eyes, nose, and mouth are used to compute an affine transformation matrix for aligning the face to a predefined template. The aligned face region is then cropped to produce a standardized face image. Both DeepfakeBench and AV-HuBERT adopt this general approach, but differ in implementation details. DeepfakeBench processes each frame independently, skipping frames where no face is detected. This may lead to shortened frame sequences and audio-video misalignment. Additionally, their cropping region is not smoothed over time, resulting in frequent jittering. In contrast, AV-HuBERT applies temporal smoothing to the cropping region by averaging landmarks across consecutive frames. However, it interpolates landmarks for frames where no face is detected, which may introduce inaccuracies that propagate into downstream detection tasks. Meanwhile, if more than one face exists in the video, cropped region will move from one face to another, leading to background region cropped sometimes. To take advantage of both approaches, we adopt a hybrid

strategy. Specifically, we discard frames with no detected face or with multiple faces to ensure clear speaker-face correspondence. Once the expected number of consecutive valid frames is collected, they are grouped together as a preprocessed video clip.

Notably, leading silence problems may occur in fake audio data. Based on statistical analysis in (Smeu et al., 2025), 25-30 ms leading silence is found in FakeAVCeleb. To avoid potential overfitting on leading silence data, we skip two frames (80ms) on each dataset during current preprocessing step.

**Preprocessed Data Storage**   Frequent data loading during training, evaluation, and analysis can become a bottleneck, especially when using MP4 files, which are storage-efficient due to encoding but incur high decoding overhead. To address this, we decode the MP4 data into ndarray format and store it using NumPy's compressed NPZ format. This approach significantly improves data loading speed but comes at the cost of increased storage usage—each clip requires approximately 1.8MB of disk space—posing challenges for I/O throughput and storage capacity, particularly in large-batch scenarios. To balance efficiency and resource constraints, we provide both MP4 and NPZ formats, allowing users to select the most suitable format based on their hardware capabilities and data loading requirements.

### B.2   Implementation Details of Multimodal Deepfake Detectors

We have implemented 11 multimodal deepfake detectors from four types of models, including vanilla baseline models, regular models, ensemble models, and pre-trained multimodal large language models. To ensure fair and consistent evaluation, all experiments are conducted in a standardized environment using 4 Nvidia RTX 3090 GPUs, each with 24GB of memory. The total training time is around 2000 GPU hours.

For regular models, we implement 7 detectors, including MDS (Chugh et al., 2020), AVTS (Sung et al., 2023), AVAD (Feng et al., 2023a), MRDF (Zou et al., 2024), and AVFF (Oorloff et al., 2024), FRADE (Nie et al., 2024), AVH (Smeu et al., 2025). For the ensemble models, we implement one model that has the same backbones as AVTS. Each ensemble combines the predictions from one visual and one audio deepfake detector using an averaging strategy. For pre-trained multimodal large language models, we use two as detectors, including Qwen2.5-Omni (Xu et al., 2025) and VideoLLaMA2 (Cheng et al., 2024). Additionally, during training, we employ loss weighting to mitigate the imbalance between real and fake samples in the dataset. Specifically, we set the weight as the ratio of real to fake samples. For example, the weight is 0.09 for Mega-MMDF.

**MDS**   MDS (Chugh et al., 2020) is a detector with dual backbones extracting video and audio features respectively. It is trained using two cross-entropy losses and one specific contrastive loss. We train with a batch size of 32 using the Adam optimizer with $beta_1 = 0.9$, $beta_2 = 0.999$, and a learning rate of $1 \times 10^{-3}$ for 10 epochs. The training time is around 5 GPU hours per epoch when trained on our Mega-MMDF.

**AVTS**   AVTS (Sung et al., 2023) is a self-supervised audio-visual mutual learning framework for deepfake detection. It leverages temporal synchronization between facial movements and speech to identify inconsistencies caused by forgeries. The model is trained in two phases: First, it learns rich spatio-temporal representations from real videos using an audio-visual temporal synchronization task without manual labels. Second, it freezes the pretrained feature extractors and trains a classifier to detect manipulations based on inconsistencies between audio and visual cues. This approach enables robust detection of both seen and unseen deepfakes, demonstrating strong generalization ability across multiple datasets and manipulation types. As the code for this model is partially open-sourced, we reproduce the training process to the best of our ability. We follow the original training paradigm described in the paper. In the first stage, we directly utilize the checkpoint provided by the authors. In the second stage, we train with a batch size of 32 using the Adam optimizer with $beta_1 = 0.9$, $beta_2 = 0.999$, and a learning rate of $2 \times 10^{-4}$ for 50 epochs with early stopping. The training time is around 1 GPU hour per epoch when trained on our Mega-MMDF.

**AVAD**   AVAD (Feng et al., 2023a) is a self-supervised audio-visual anomaly detection framework for deepfake forensics. It models the natural synchronization patterns between speech and facial movements using only real, unlabeled videos, and identifies forgeries by detecting anomalies in these

patterns. The framework first leverages a pretrained audio-visual synchronization network to extract temporal alignment features, such as time delay distributions between audio and lip motion. Then, an autoregressive Transformer is trained to capture the probability distribution of these synchronization features. During inference, sequences with low likelihood are flagged as manipulated, since deepfakes often disturb subtle audio-visual coherence. In our reproduction, we follow the training settings described in the original paper. In the first stage, the audio-visual synchronization model is trained on real speech video datasets in our Mega-MMDF. We train with a batch size of 48 using the Adam optimizer with $beta_1 = 0.95$, $beta_2 = 0.999$, and a learning rate of $1 \times 10^{-4}$ for 20 epochs. In the second stage, the extracted synchronization features are used to train an anomaly detection model. We train with a batch size of 64 using the Adam optimizer with $1 \times 10^{-3}$ learning rate, weight decay of $1 \times 10^{-6}$ and warm-up and cosine learning rate decay strategies for 20 epochs. The training time is around 5 GPU hours per epoch when trained on our Mega-MMDF for the first stage, and 6 GPU hours per epoch for the second stage.

**MRDF**  MRDF (Zou et al., 2024) leverages both audio and visual streams to detect deepfakes more robustly. The approach incorporates a two-stream network architecture, where audio and visual features are extracted separately using pre-trained backbones and then fused to perform joint classification. To enhance performance, the framework integrates a cross-modal consistency module that encourages alignment between modalities, improving the model's ability to detect subtle inconsistencies in manipulated content. During our training process, we strictly follow the experimental settings described in the original paper. We train with a batch size of 64 using the Adam optimizer with $beta_1 = 0.5$, $beta_2 = 0.9$, and a learning rate of $1 \times 10^{-3}$ for 30 epochs with early stopping. The training time is around 3.5 GPU hours per epoch when trained on our Mega-MMDF.

**AVFF**  AVFF (Oorloff et al., 2024) integrates audio-visual features to detect fake videos through a two-stage multimodal contrastive learning framework. In the first stage, the model is trained on real videos to learn the natural correspondence between speech and facial movements by combining contrastive learning with a cross-modal reconstruction strategy. It uses a dual-encoder structure inspired by CAV-MAE to separately extract features from audio and visual inputs, which are then aligned in a shared space. In the second stage, the learned representations are used to train a supervised classifier that distinguishes real from manipulated videos. This design enables the model to effectively capture temporal dynamics and inconsistencies across modalities, making it highly effective for deepfake detection. We follow the original training paradigm described in the paper during our training process. In the first stage, we train with a batch size of 48 using the Adam optimizer with $beta_1 = 0.95$, $beta_2 = 0.999$, and a learning rate of $1 \times 10^{-4}$ for 50 epochs. In the second stage, we train with a batch size of 64 using the Adam optimizer with $beta_1 = 0.95$, $beta_2 = 0.999$, and a learning rate of $1 \times 10^{-5}$ for 20 epochs. The training time is around 2.5 GPU hours per epoch when trained on our Mega-MMDF for the first stage, and 8.5 GPU hours per epoch for the second stage.

**FRADE**  FRADE (Nie et al., 2024) is an adapter-designed network based on ViT (Dosovitskiy et al., 2021). Its AFI adapter is to extract features within the frequency domain, while the ACI adapter is to mix visual features with audio features on each layer. The official code does not contain training code. We train with a batch size of 32 using the AdamW optimizer with $beta_1 = 0.9$, $beta_2 = 0.999$, and a learning rate of $2 \times 10^{-6}$ for 10 epochs. The training time is around 14 GPU hours per epoch when trained on our Mega-MMDF.

**AVH**  AVH (Smeu et al., 2025) is a network using pretrained AV-HuBERT (Shi et al., 2022) to extract unimodal features. 4 layers of MLP are used to align the feature dimensions. In this work, an unsupervised approach and a supervised one are mentioned. Since we do not have adjacent 15 video frames available for each audio frame, we only implement the supervised approach. Following the same settings used in the official code, we implement this method by concatenating networks together instead of saving features first and training the MLP second in the official code. We train with a batch size of 32 using the Adam optimizer with $beta_1 = 0.9$, $beta_2 = 0.999$, and a learning rate of $1 \times 10^{-3}$ for 10 epochs. The training time is around 8 GPU hours per epoch when trained on our Mega-MMDF.

**Vanilla Baseline**    Our vanilla baseline follows a naive two-stage training pipeline consisting of a visual backbone, an audio backbone, and an MLP classifier. In the first stage, we extract visual and audio features using the respective backbones and train with a contrastive loss to align the modalities. In the second stage, we freeze both backbones and train the MLP classifier. Specifically, in the first stage, we train on real samples from the Mega-MMDF training set for 20 epochs using the Adam optimizer with a learning rate of $4 \times 10^{-4}$. In the second stage, we also use the Adam optimizer with a learning rate of $1 \times 10^{-4}$ and train for 30 epochs with early stopping. The training time is around 2 GPU hours per epoch when trained on our Mega-MMDF for the first stage, and 3 GPU hours per epoch for the second stage.

**Ensemble**    The ensemble method directly averages the audio and visual detection results from the single-modality detectors. Both detectors are trained on their respective forgery data without any cross-modality information interaction. This ensemble strategy allows each model to only focus on single-modality forgery artifacts, serving as a straightforward baseline for multimodal deepfake detection. Specifically, we leverage a video backbone C3D-ResNet18 as the visual detector and SE-ResNet18 as the audio detector. In our experiments, we train our visual detector with a batch size of $48$ using the Adam optimizer with $weight\_decay = 0.01$ and a learning rate of $3 \times 10^{-4}$ for 2 epochs. Besides, we train our audio detector with a batch size of 32 using the Adam optimizer with a learning rate of $3 \times 10^{-4}$ for 4 epochs. The training time for C3D-ResNet18 is around 4.5 GPU hours per epoch when trained on our Mega-MMDF. And the training time for SE-ResNet18 is around 5 GPU hours per epoch.

**Qwen2.5-Omni**    Qwen2.5-Omni is an end-to-end multimodal foundation model capable of understanding and processing diverse input types such as text, images, audio, and video, while generating both text and natural speech responses in real time. It uses a unified streaming architecture with synchronized audio-visual encoding and a novel position embedding method to align temporal information across modalities. Its robust cross-modal representation and generation capabilities offer a strong foundation for identifying inconsistencies and manipulations across different media types. During the experiment, we use its default configuration. For multimodal deepfake detection, we use the following prompt: "Carefully analyze the video and audio to determine whether this audiovisual content is real or fake. Provide only a single-word response: Real or Fake." During inference, we extract the model's output token probabilities for the words "Real" and "Fake," and use these probabilities to compute the AUC metric. The total inference time on Mega-MMDF's test set is around 5 hours.

**VideoLLaMA2**    VideoLLaMA2 is an open-source video-language model designed to improve the understanding of both spatial-temporal video content and audio information. It introduces a specialized spatial-temporal convolution module to better capture dynamic visual patterns and incorporates audio features through a dedicated training branch, enabling richer multimodal reasoning. Its ability to model both visual and auditory signals can be used to identify subtle cross-modal inconsistencies indicative of manipulated content. During the experiment, we use VideoLLaMA2 in the default configuration. For multimodal deepfake detection, the prompt is: "Carefully analyze the video and audio to determine whether this audiovisual content is real or fake. Provide only a single-word response: Real or Fake." During inference, we extract the model's output token probabilities for the words "Real" and "Fake," and use these probabilities to compute the AUC metric. The total inference time on Mega-MMDF's test set is around 5 hours.

### B.3    DETAILS OF CODEBASE

**Data Preprocessing Module** This module standardizes data preprocessing across all supported datasets to eliminate cross-dataset heterogeneity (*e.g.*, variations in sampling specifications and unaligned face regions). Synchronized audio-video clips are generated through three sequential submodules: *Clip Splitting*, *Assembling*, and *Transcoding*. **(1) Clip Splitting**: This submodule provides utilities to process raw audiovisual data with user-defined hyperparameters (*e.g.*, clip length, video frame rate, audio sampling rate, and number of clips). All files are preprocessed through a standardized pipeline including: (i) audio-video stream separation, (ii) audio sampling and video frame rate adjustment to address heterogeneity in sampling specifications, and (iii) face alignment and cropping to ensure spatial consistency and extract consecutive frames with well-aligned faces. The

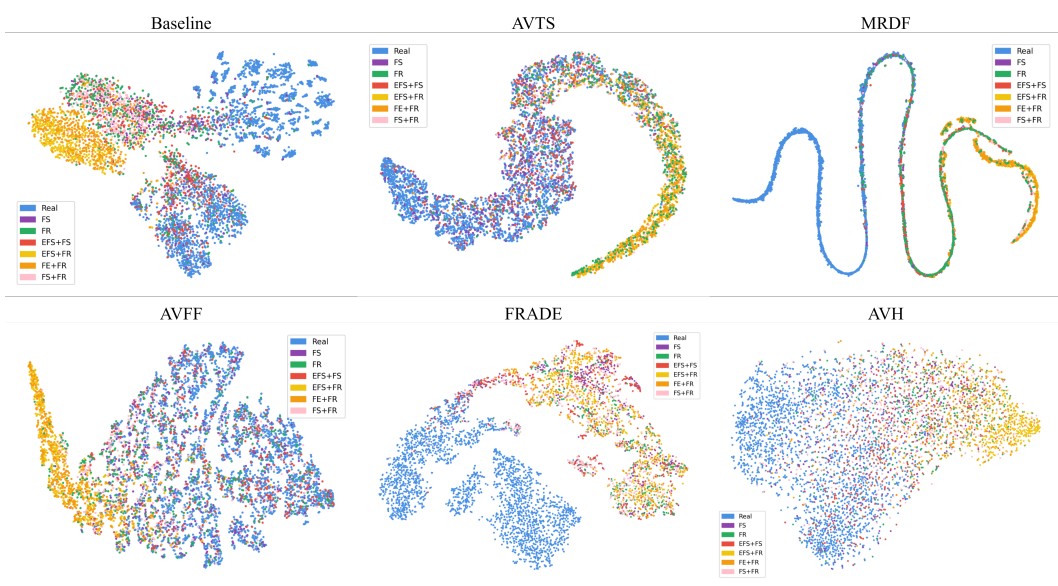

Figure 5: t-SNE visualization of latent features from six different models trained and tested on our Mega-MMDF dataset. The models include Baseline, AVTS, MRDF, AVFF, FRADE, and AVH.

processed outputs are stored in `WAV` (audio) and `MP4` (video) formats, respectively. **(2) Assembling**: This submodule processes preprocessed audio/video clips by pairing each clip with its corresponding metadata, label, and the partition information from the original dataset. For each paired data instance, a standardized `JSON` entry is constructed, with all entries aggregated into a unified `JSON` file. **(3) Transcoding**: This submodule provides an optional decoding process for preprocessed `MP4` clips, extracting frames and storing them in binary-encoded `NPZ` files to mitigate video reading bottlenecks. This optimization trades increased storage requirements and higher disk I/O pressure for improved loading speed, allowing users to balance performance and storage efficiency based on their hardware capabilities.

**Detector Module** It supports: **(1) Loading of backbones**, including various widely used backbone architectures (*e.g.*, SEResNet (Hu et al., 2018), C3D-ResNet18 (Tran et al., 2015)) with pretrained weights; **(2) Customization of new architecture**, which enables flexible design of detector architectures by integrating backbone networks with user-defined blocks.

**Training Module** This module supports both **one-stage** and **multi-stage** training paradigms, providing a standardized protocol that includes optimizer configuration, learning rate scheduling, initialization of learning rates and model weights, epoch management, and distributed training setup. Additionally, it implements a training-with-validation paradigm to automatically select the best-performing checkpoint based on validation set performance, ensuring optimal generalization.

**Evaluation and Analysis Module** **(1) Evaluation**: This submodule defines multiple evaluation protocols (detailed in the next section) to assess detectors, including a comprehensive set of quantitative metrics (*e.g.*, Accuracy (ACC), Area Under the Curve (AUC), Average Precision (AP), and Equal Error Rate (EER)) and visualizations (*e.g.*, Receiver Operating Characteristic (ROC) and Precision-Recall (PR) curves) for clearer interpretation of results. **(2) Analysis**: This submodule provides analytical tools for in-depth investigation, such as *t-SNE Visualization* (Van der Maaten & Hinton, 2008) to project high-dimensional features into a low-dimensional space, revealing feature-space data structure; *Prediction Score Histogram* for comparing model confidence between real and fake samples; and *Audio Spectrograms* and *Image Frequency Maps* to highlight forgery artifacts in audio and visual modalities.

### B.4 MORE ANALYSIS OF RESULTS

**Fairness** *Multimodal detectors trained on our balanced dataset exhibit fair performance across gender and skin tone.* Figure 6 visualizes per-subgroup AUC scores for detectors trained on our Mega-MMDF, along with two zero-shot multimodal large language models (MLLMs). Overall, the regular detectors trained on Mega-MMDF demonstrate balanced performance across gender and skin tone subgroups, suggesting that our dataset's demographic balance effectively mitigates common biases. In contrast, zero-shot MLLMs show mixed results: their subgroup performance appears more influenced by pretraining data. For instance, Qwen2.5-Omni achieves relatively balanced detection performance, whereas VideoLLaMA2 reveals subgroup-specific biases. These observations underscore the importance of training data composition in building fair and generalizable multimodal detectors.

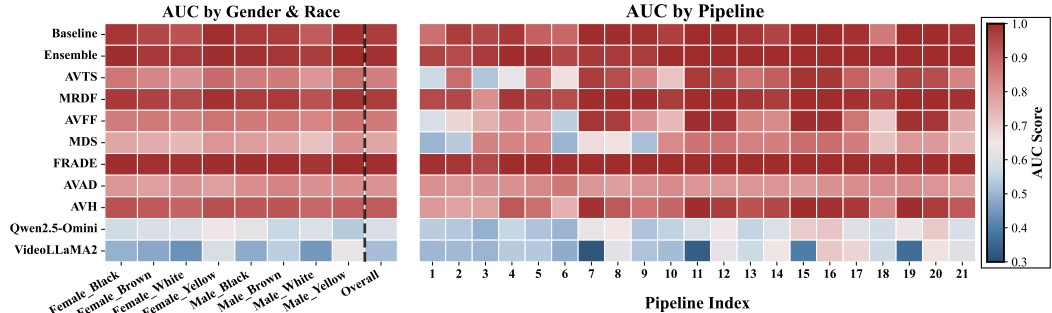

Figure 6: **Left:** Performance of detectors across different subgroups, trained and tested on our Mega-MMDF. **Right:** Performance across different forgery pipelines, trained and tested on our Mega-MMDF.

**Per-pipeline Performance Breakdown** *Detectors' intra-dataset performance on Mega-MMDF is unbalanced.* With 21 forgery pipelines spanning 7 forgery types, Mega-MMDF provides a fine-grained lens into detector behavior under diverse manipulations. Figure 6 presents the per-pipeline analysis. VideoLLaMA2 shows pronounced weaknesses on Pipelines 7, 11, 15, and 19, all involving EFS (Entire Face Synthesis), likely due to limited prior exposure to such forgeries. In contrast, it performs better on Pipelines 8, 12, 16, and 20, which involve FE (Face Editing). This discrepancy is notable: despite FE altering smaller regions than EFS, VideoLLaMA2 appears more sensitive to boundary-level inconsistencies introduced by FE. MDS experiences performance drops on Pipelines 1, 2, 6, 7, 8, and 9, all of which contain real audio, suggesting that it is more effective when both modalities are forged. By comparison, other detectors exhibit consistently strong results on pipelines involving EFS and FE, implying that these artifacts are relatively easier to learn. This observation is consistent with Analysis 2, where stacked forgeries (*e.g.*, EFS+FR) are largely shaped by EFS artifacts, underscoring detectors' tendency to overfit to the most salient cues.

## C LLM USAGE

Large Language Models (LLMs) are used in the writing of this paper solely for correcting grammar and improving the clarity of sentences. No part of the research design, experiments, analysis, or results rely on LLMs.

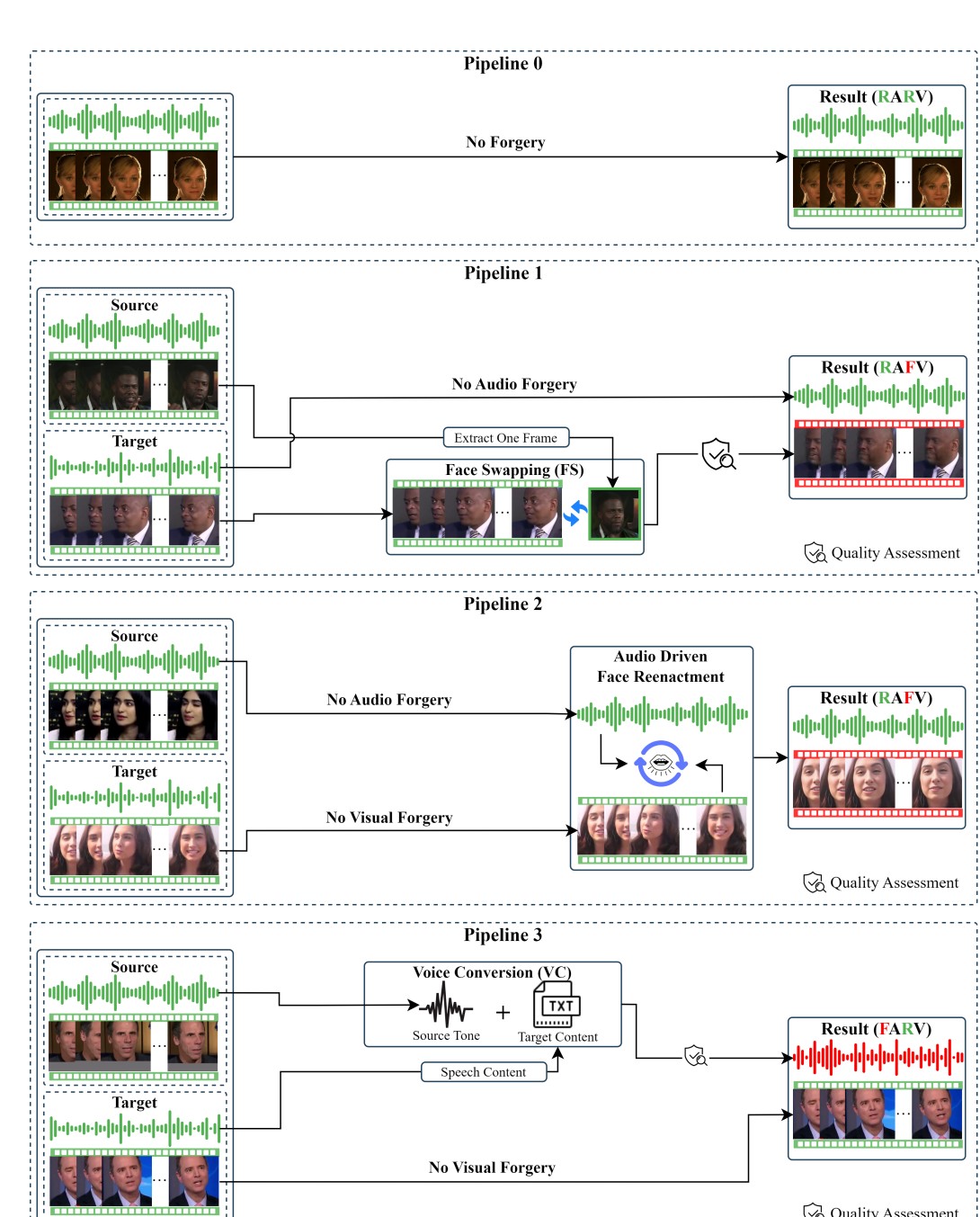

Figure 7: Illustration of Pipelines 0 to 3.

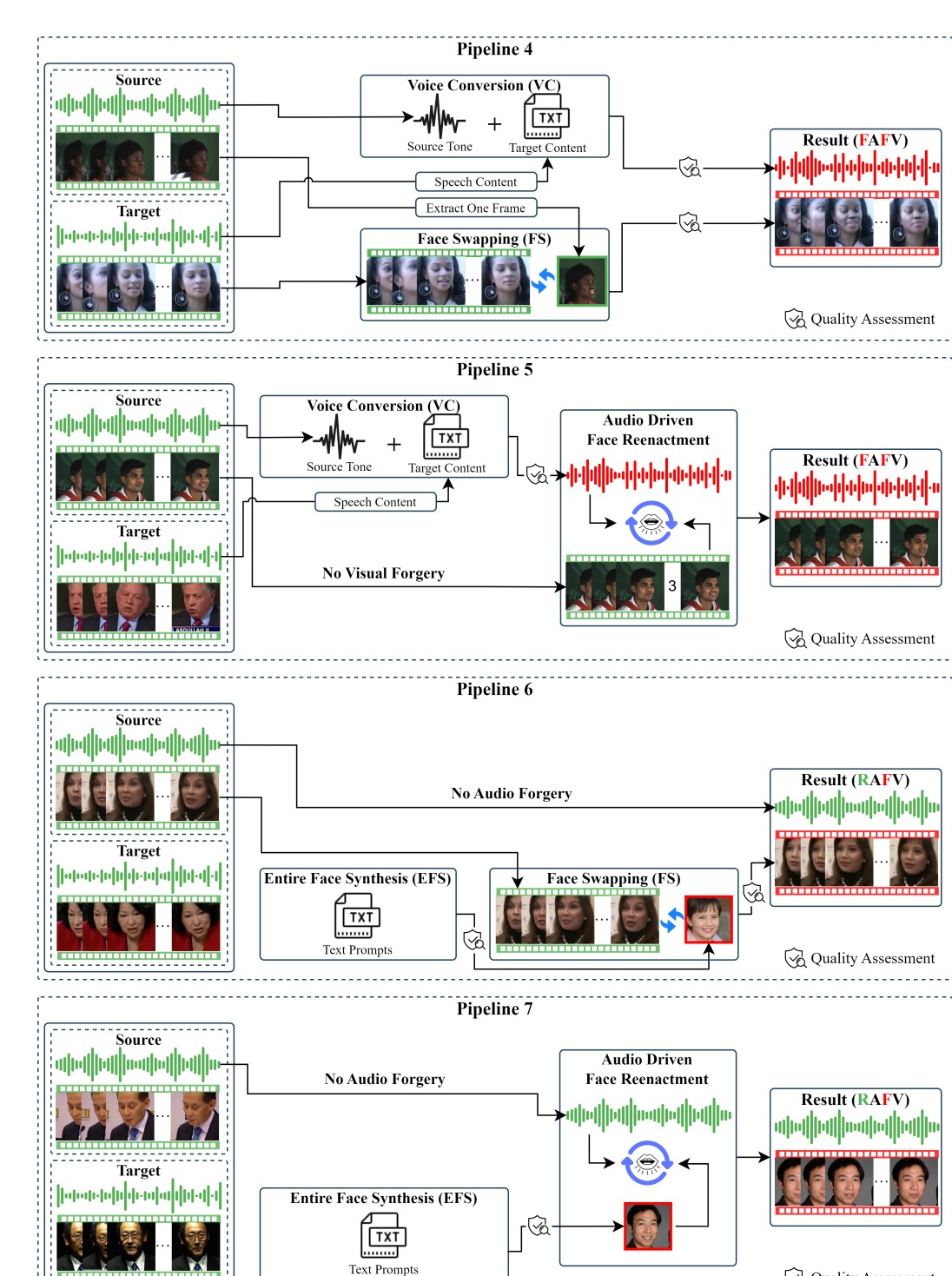

Figure 8: Illustration of Pipelines 4 to 7.

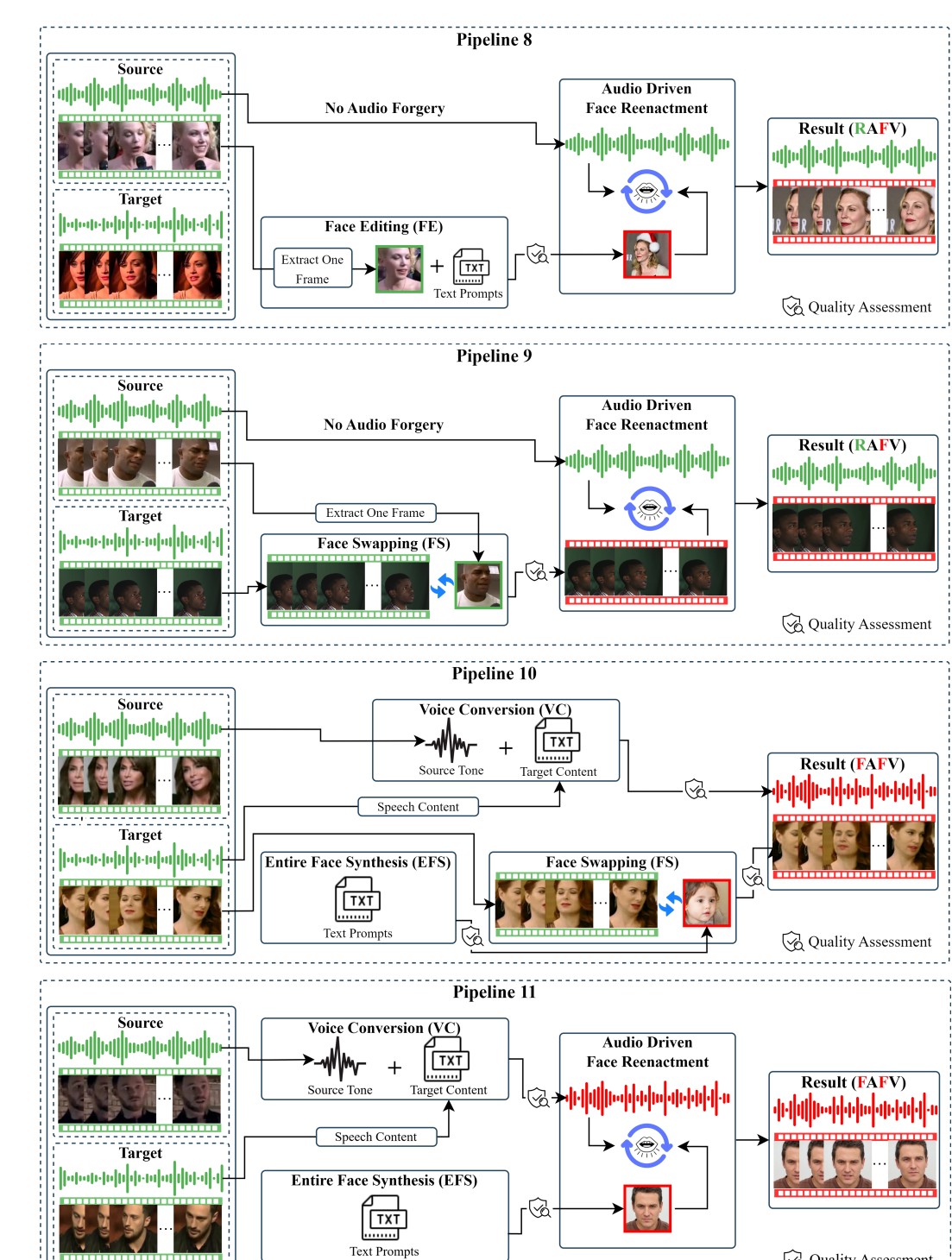

Figure 9: Illustration of Pipelines 8 to 11.

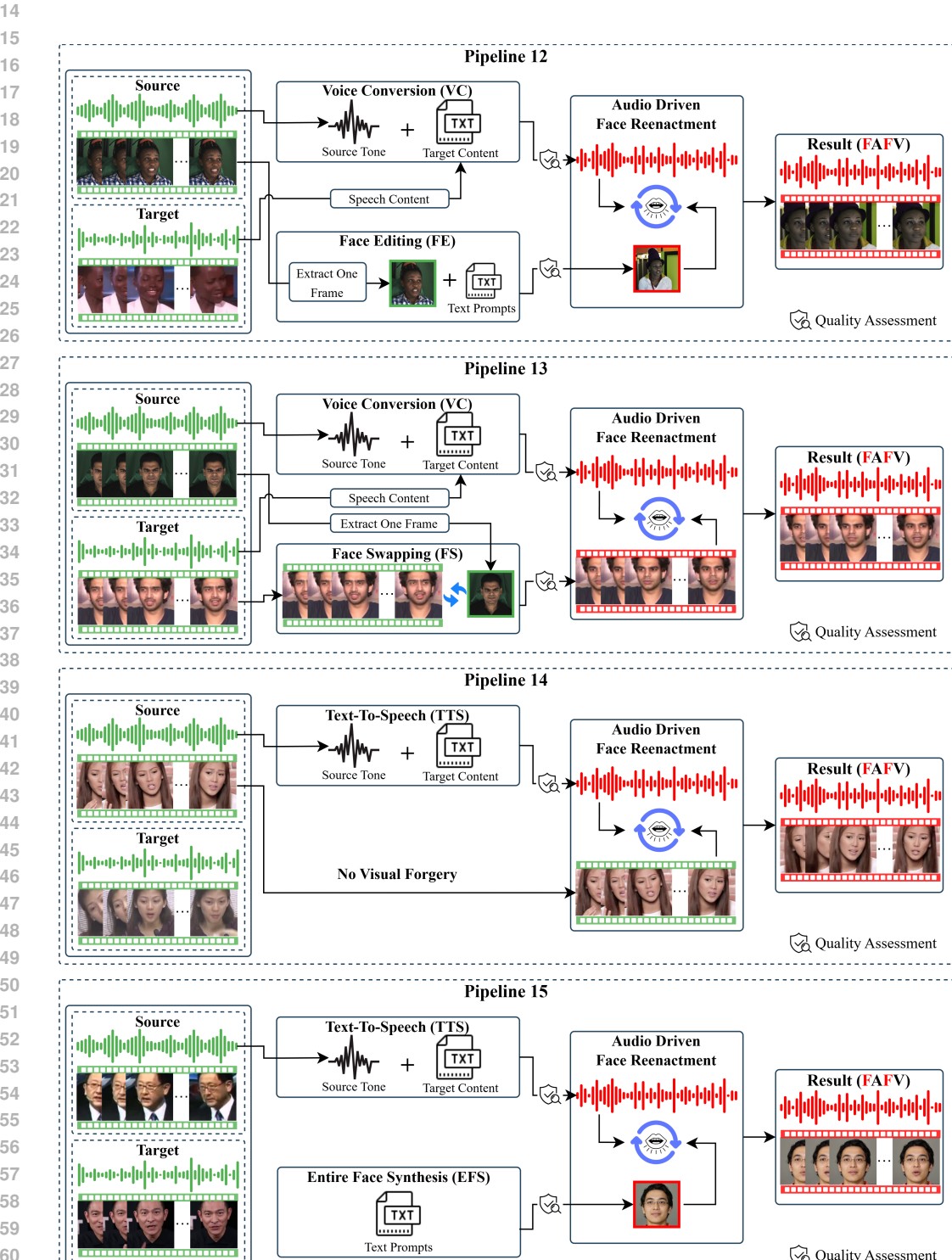

Figure 10: Illustration of Pipelines 12 to 15.

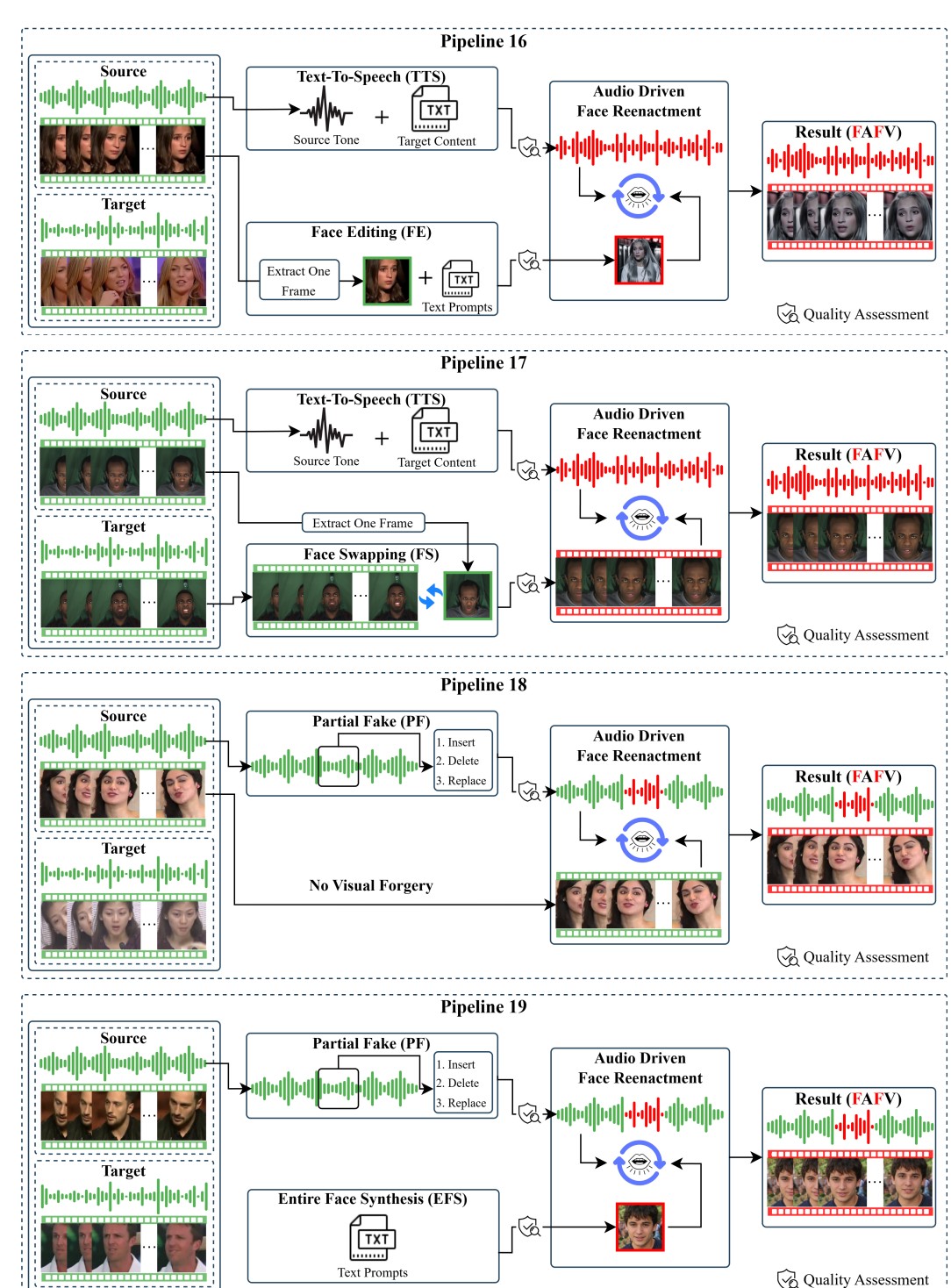

Figure 11: Illustration of Pipelines 16 to 19.

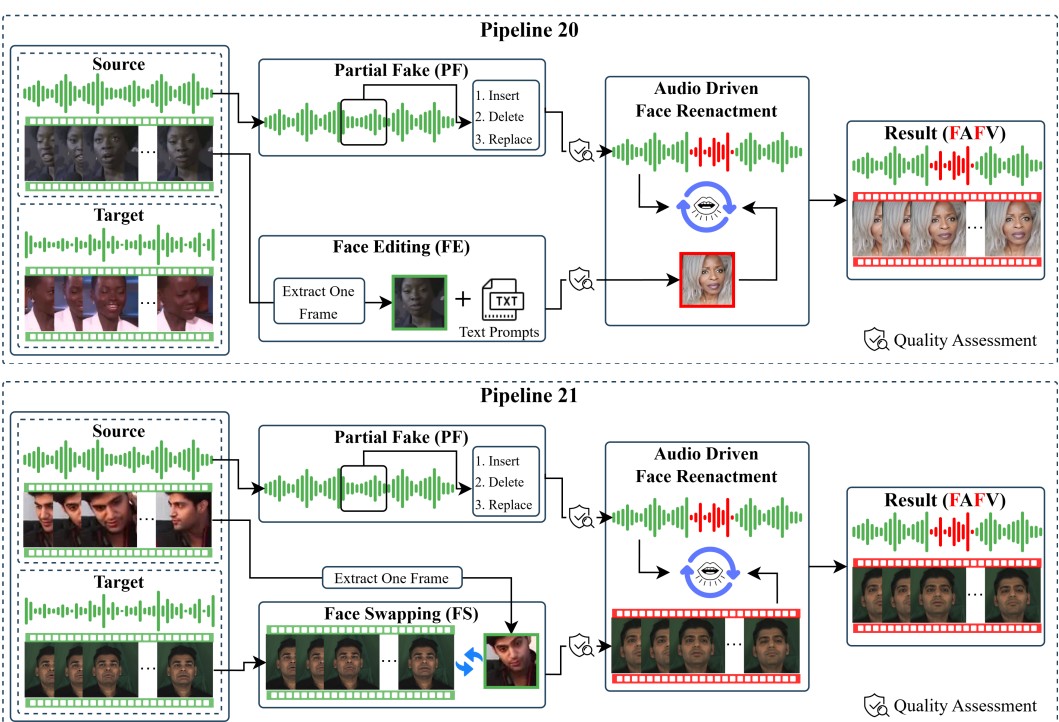

Figure 12: Illustration of Pipelines 20 to 21.

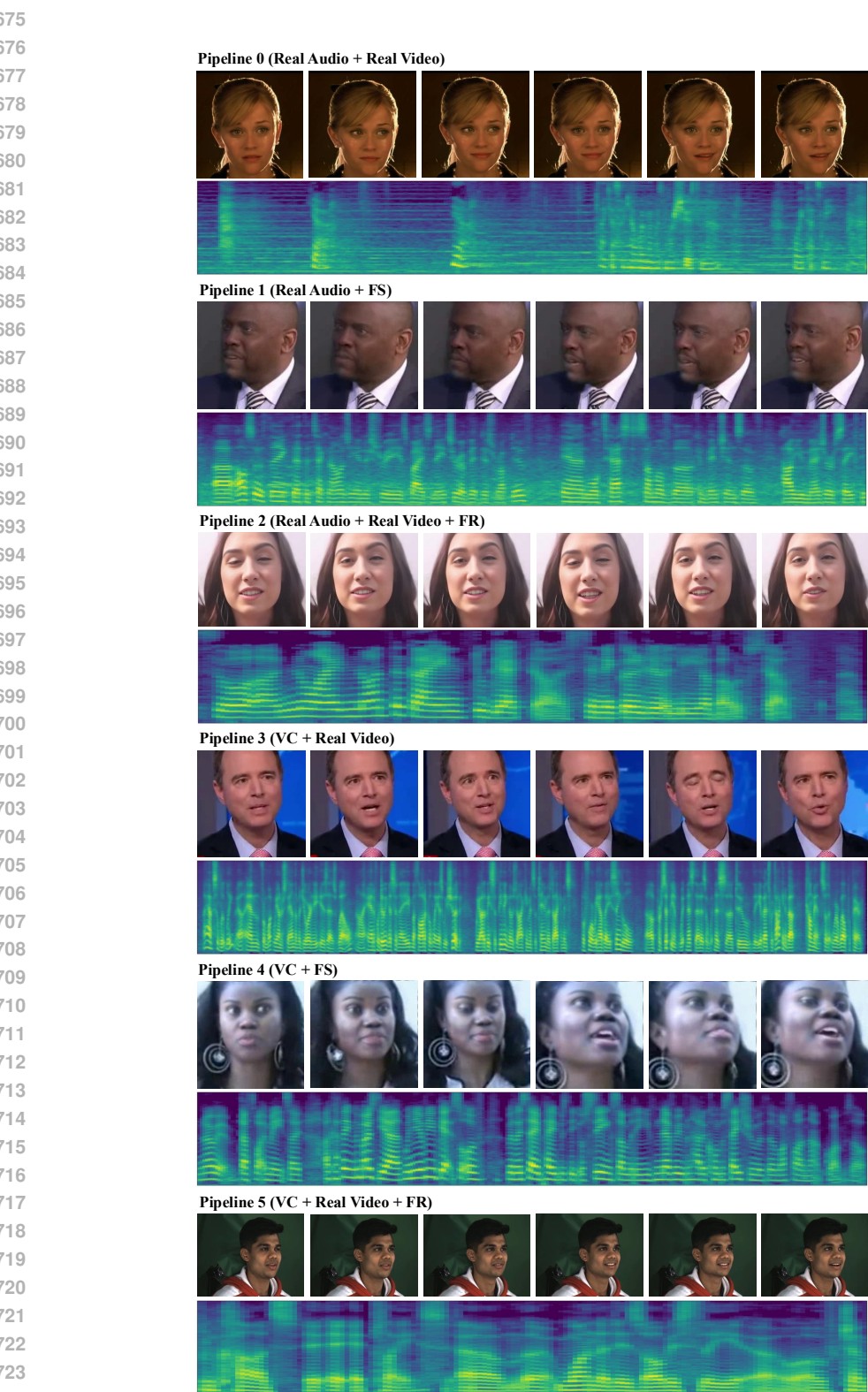

Figure 13: Visualization of selected samples from our Mega-MMDF dataset.

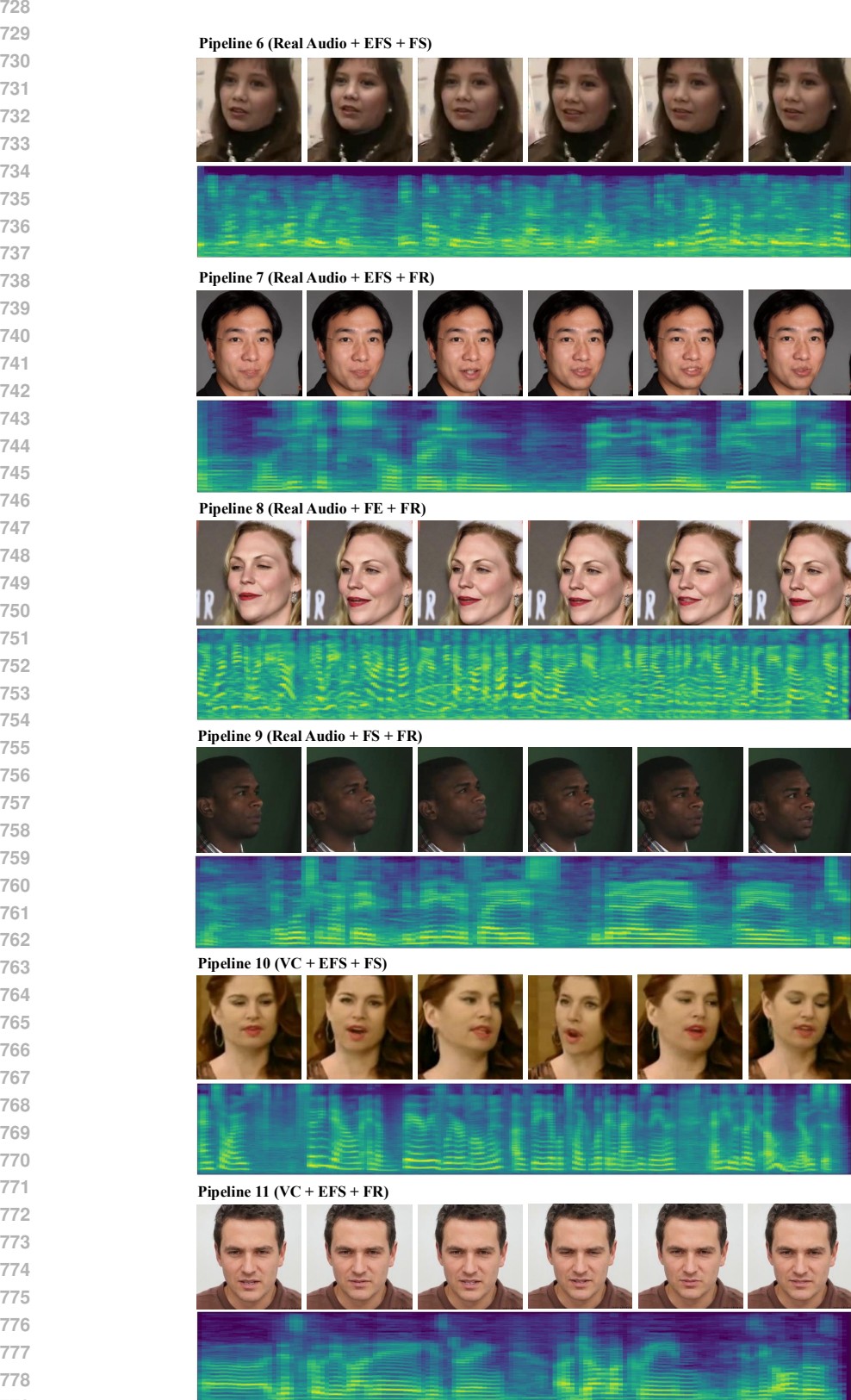

Figure 14: Visualization of selected samples from our Mega-MMDF dataset.

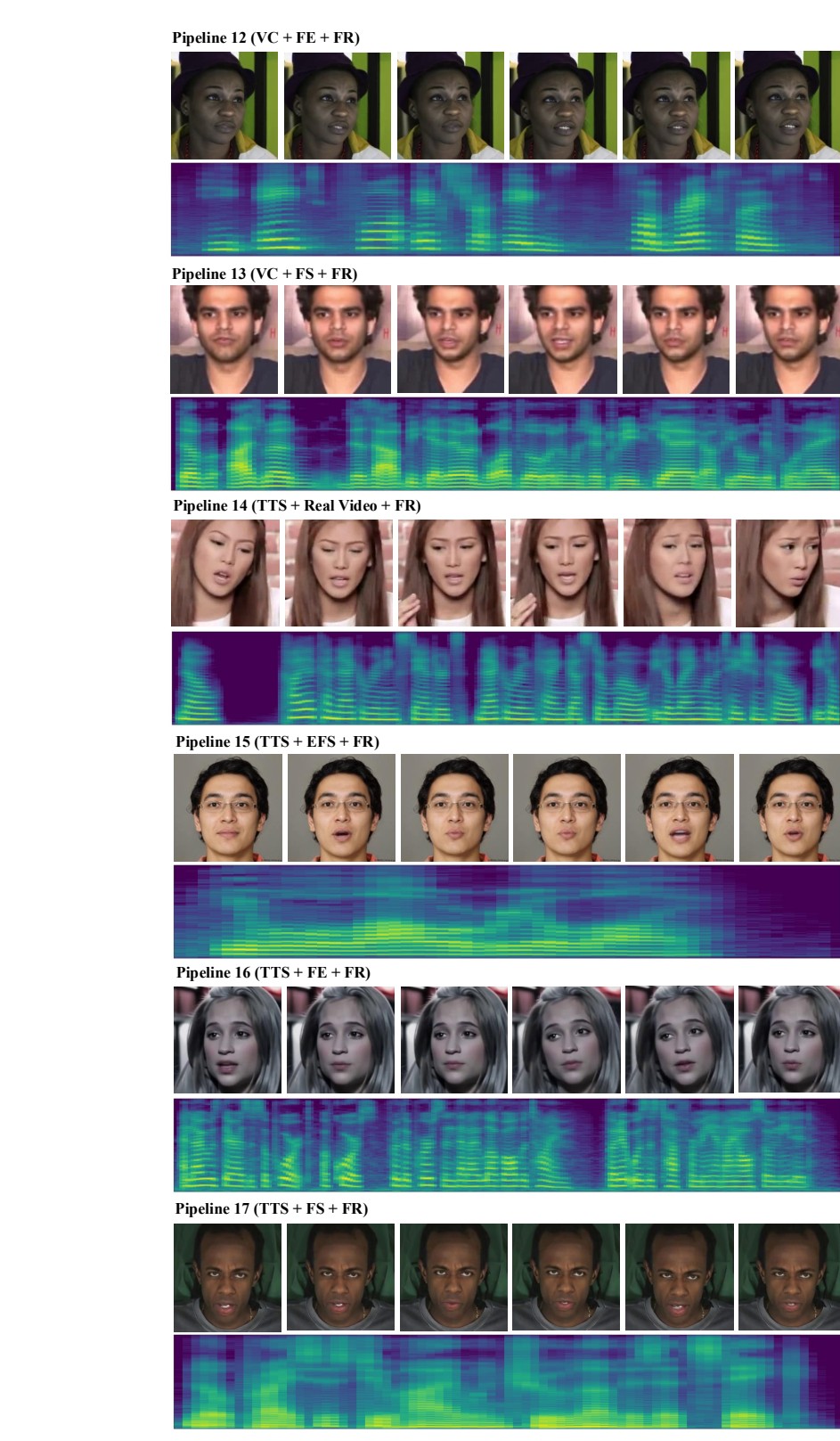

Figure 15: Visualization of selected samples from our Mega-MMDF dataset.

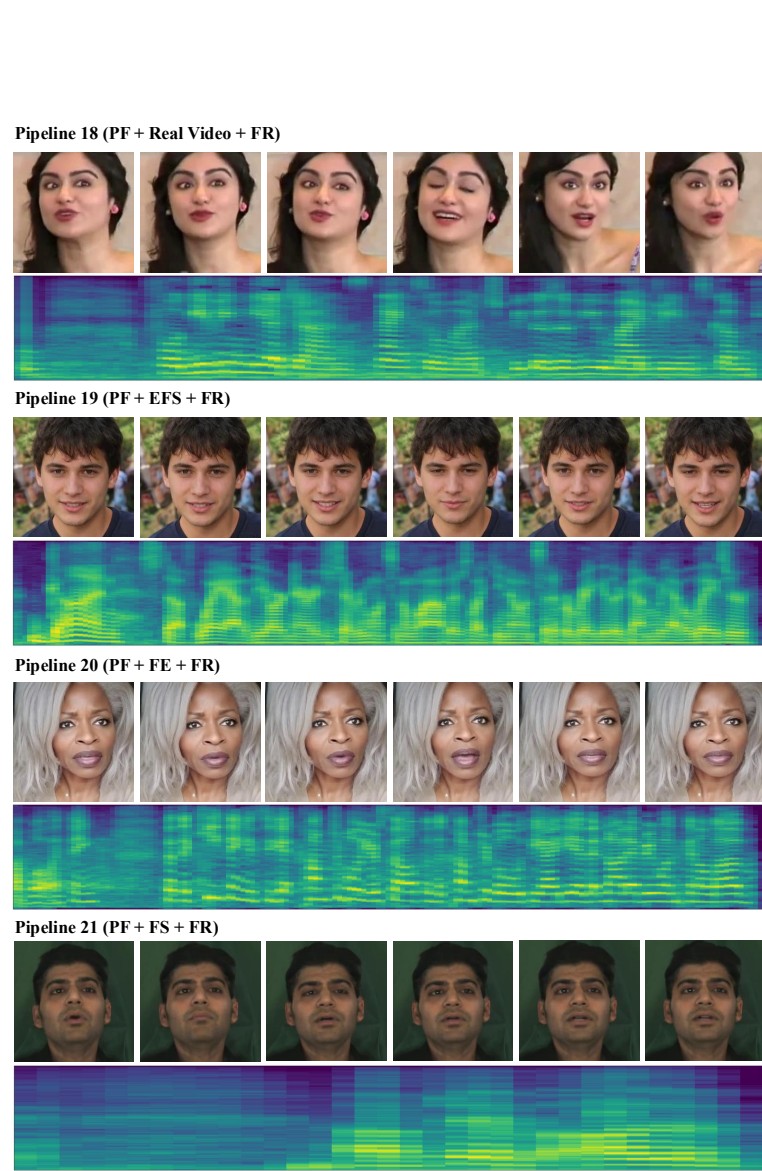

Figure 16: Visualization of selected samples from our Mega-MMDF dataset.

