# OpenReview forum: "DeepfakeBench-MM: A Comprehensive Benchmark for Multimodal Deepfake Detection"
_ICLR.cc/2026/Conference — ICLR 2026 Conference Withdrawn Submission_

### Official Review · Reviewer_x6ca · 2025-10-27

**Soundness:** 3
**Presentation:** 3
**Contribution:** 3
**Rating:** 6
**Confidence:** 4

**Summary:**

The manuscript builds a large-scale, diverse, and high-quality dataset for multimodal deepfake detection and proposes the first unified benchmark for multimodal deepfake detection, advancing the foundational infrastructure for multimodal deepfake detection.

**Strengths:**

The manuscript constructs a large-scale multimodal deepfake dataset and proposes a multimodal deepfake detection benchmark, advancing the development of multimodal detection. It is well-organized and clearly presents the limitations of existing research and the strengths of the proposed approach.

**Weaknesses:**

(1) The authors need to provide a more detailed description of the content in Figure 1 to better highlight the advantages of the proposed dataset.
(2) The authors should clarify how the thresholds for each metric in Section 3.3 were determined.
(3) In Section 3.3, the STT model WhisperX is used to evaluate audio fidelity. What specific metric is employed for this assessment?
(4) [1] and [2] also propose ensemble approaches, how does the ensemble model in Section 4.1 differ from theirs?
(5) In Analysis 2 of Section 4.3, how is it determined that EFS remains dominant in combinations? Why were no experiments conducted with EFS-only?
[1] Zhixi Cai, Shreya Ghosh, Aman Pankaj Adatia, Munawar Hayat, Abhinav Dhall, Tom Gedeon, and Kalin Stefanov. Av-deepfake1m: A large-scale llm-driven audio-visual deepfake dataset. In ACM International Conference on Multimedia, 2024.
[2] Kartik Thakral, Rishabh Ranjan, Akanksha Singh, Akshat Jain, Richa Singh, and Mayank Vatsa. ILLUSION: Unveiling truth with a comprehensive multi-modal, multi-lingual deepfake dataset. In International Conference on Learning Representations, 2025.

**Questions:**

(1) The authors need to provide a more detailed description of the content in Figure 1 to better highlight the advantages of the proposed dataset.
(2) The authors should clarify how the thresholds for each metric in Section 3.3 were determined.
(3) In Section 3.3, the STT model WhisperX is used to evaluate audio fidelity. What specific metric is employed for this assessment?
(4) [1] and [2] also propose ensemble approaches, how does the ensemble model in Section 4.1 differ from theirs?
(5) In Analysis 2 of Section 4.3, how is it determined that EFS remains dominant in combinations? Why were no experiments conducted with EFS-only?
[1] Zhixi Cai, Shreya Ghosh, Aman Pankaj Adatia, Munawar Hayat, Abhinav Dhall, Tom Gedeon, and Kalin Stefanov. Av-deepfake1m: A large-scale llm-driven audio-visual deepfake dataset. In ACM International Conference on Multimedia, 2024.
[2] Kartik Thakral, Rishabh Ranjan, Akanksha Singh, Akshat Jain, Richa Singh, and Mayank Vatsa. ILLUSION: Unveiling truth with a comprehensive multi-modal, multi-lingual deepfake dataset. In International Conference on Learning Representations, 2025.

---

### Official Review · Reviewer_3Rtj · 2025-10-28

**Soundness:** 2
**Presentation:** 3
**Contribution:** 2
**Rating:** 4
**Confidence:** 3

**Summary:**

This paper introduces Mega-MMDF, a large-scale multimodal deepfake detection dataset, and DeepfakeBench-MM, a standardized, extensible benchmark for evaluating audio-visual deepfake detectors. The dataset is constructed via 21 compositionally diverse forgery pipelines combining 10 audio, 12 visual, and 6 face reenactment methods, resulting in 1.2 million samples. The authors present comprehensive analyses and comparisons across 5 multimodal datasets and 11 contemporary detectors, yielding insights into modality, feature fusion, generalization, and training strategies, with the intent to propel future research by providing robust infrastructure and reproducible baselines.

**Strengths:**

1. Mega-MMDF is one of the largest and most diverse multimodal deepfake datasets, substantially surpassing prior datasets in both the number of forgery methods and overall sample size.

2. The dataset construction includes an elaborate, multi-stage quality assessment for audio, video, and synchronization.

3. The evaluations are exhaustive, covering intra-dataset, cross-dataset, and cross-pipeline detection.

**Weaknesses:**

1. The paper’s primary technical contributions rest in data and benchmark construction, not in algorithmic advances or new detection paradigms. While infrastructure is critical, there is minimal advancement on detection methodology itself.

2. Although Mega-MMDF boasts scale and diversity, the potential for overfitting to known or compositional artifacts is briefly mentioned but lacks rigorous quantitative measures of “wildness” versus real-world deepfake complexity.

3. The benchmark focuses on sample-level binary detection, but current research increasingly targets fine-grained, temporally or spatially localized manipulations. While the paper includes the LAV-DF and AVDeepfake1M datasets (which focus on localization), the methodology and discussion do not explicitly address or propose extensions for evaluation in localization rather than binary settings.

4. While it provides notable unification, the statement of "the first" unified benchmark is over-strong, as paper itself references efforts such as FakeAVCeleb and ILLUSION, as well as ad hoc protocols from prior work.

**Questions:**

The major concern is the limited methodological contribution of this work.

While I fully acknowledge the importance of constructing a unified dataset for deepfake detection, the contribution presented here appears incremental, given the existence of prior efforts such as DeepfakeBench and DF40, which have already aimed to establish standardized benchmarks for this task.

---

### Official Review · Reviewer_fieR · 2025-10-31

**Soundness:** 3
**Presentation:** 1
**Contribution:** 3
**Rating:** 4
**Confidence:** 5

**Summary:**

The paper proposes a large scale diverse dataset called Mega-MMDF and DeepfakeBench-MM, which is a unified benchmark for multimodal deepfake detection. The paper unifies the pre-processing pipeline which standardizes the evaluation.

**Strengths:**

1. Lack of standardized evaluation is a long standing issue in the community. The authors are addressing a real research gap
2. The authors share the code and also mention "continuous expansion", showcasing their commitment to maintain a comprehensive Multimodal Deepfake Detection benchmark.
3. The diversity of Mega-MMDF is impressive.

**Weaknesses:**

1. The authors need to demonstrate that training on Mega-MMDF leads to better numbers on all test sets. The paper is missing performance comparison where models are trained on other recent train sets. I am still not convinced that the quality of the train set is good enough. I am not able to conclude anything from Table 2.

2. The authors should include a failure case analysis to showcase the model trained using Mega-MMDF fails in which cases.

3. The models trained on Mega-MMDF performs good on intra-dataset evaluation, however the cross-dataset performance seems poor. The authors mention reasons in Line 321 - 323 as to why the model trained on Mega-MMDF fails on those datasets. However, I think it is a major weakness as the train dataset proposed lacks those variations. With a project this ambitious, I expect that the model trained on it should improve on all cross-dataset evaluation scenario.

4. The organization and the way of presenting the experiments (results and tables) can be improved. It takes multiple reads to actually understand.

**Questions:**

The questions are mentioned in the Weakness Section.

---

### Official Review · Reviewer_U3bf · 2025-11-02

**Soundness:** 2
**Presentation:** 3
**Contribution:** 2
**Rating:** 2
**Confidence:** 5

**Summary:**

The authors present a significant engineering effort comprising two main components: the Mega-MMDF dataset and the DeepfakeBench-MM framework. The Mega-MMDF dataset is a new, large-scale (1.1 million fake samples) multimodal (Audio-Visual) deepfake dataset, and the DeepfakeBench-MM is a unified benchmark platform designed to standardize the evaluation of multimodal deepfake detectors. The authors use this benchmark to evaluate 11 existing detectors across 5 datasets. Additionally, the paper presents several analyses based on their benchmark results.

**Strengths:**

1.	The primary strength is the creation and open-sourcing of Mega-MMDF. A dataset of this scale (1.1M fake samples) and documented diversity (21 pipelines) is a substantial contribution that will undoubtedly fuel future research.

2.	The multimodal deepfake field lacks standardized evaluation before. Author’s DeepfakeBench-MM provides a much-needed, unified, and extensible platform for fair comparison, which is critical for measuring real progress. The benchmarking of 11 detectors across 5 datasets is a thorough and valuable piece of work.

**Weaknesses:**

1.	Authors slightly hide their dataset outerlink in the anonymous github, and linked page just reveal author’s information including names, university, and the fact that this paper was double-submitted to NIPS benchmark and dataset. : https://dataverse.harvard.edu/dataset.xhtml?persistentId=doi:10.7910/DVN/J4DVAA&#41

2.	The paper's "key findings" (Sec 4.3) are presented as novel contributions, but they are largely well-known already.  For instance, Analysis 4 (Modality Bias) is a widely documented issue in multimodal learning (e.g., in AV-VQA). The authors fail to demonstrate what makes this finding unique or surprising specifically for deepfake detection. Similarly, Analysis 3 (Finetuning) simply confirms that training more parameters leads to better in-domain performance, which is not an insight. In summary, it stops short of a deep investigation.

3.	The authors did not assess the quality measurement for LAV-DF and AVDeepfake1M in purpose. This shortage highly damage the author’s argument that their dataset is more realistic than others.

**Questions:**

1.	Why didn’t you show the quality difference between AVDeepfake1M and your new dataset? (Table 1)

2.	Did the authors test if models finetuned on Mega-MMDF show worse or better generalization to other datasets? For the more importantly, did you validate the finetuned models to unseen, in-the-wild deepfake types not present in your dataset?

3.	Giving the paper’s acknowledgement about preprocessing (line 258), how the authors enforces audio-visual synchronization? Since the authors explain that they opt for a hard-coded “skip ttwo frames” approach before feeding the data to the detectors, this is the naive approach simply masking one specific artifact while ignoring the broader problem of A/V desynchronization?

4.	What’s the reason of showing only two models’ result in the Analysis 5, even though the paper assess 11 detectors for the benchmark?

**Details Of Ethics Concerns:**

It was easy to find the dataset release in other website, since the authors did not remove it. :https://dataverse.harvard.edu/dataset.xhtml?persistentId=doi:10.7910/DVN/J4DVAA&#41
The website tells the author’s name. The author should delete the link in their ReadMe.md from their anonymous github.

---

### Note · Authors · 2025-11-14

I have read and agree with the venue's withdrawal policy on behalf of myself and my co-authors.